# Better Neural PDE Solvers Through Data-Free Mesh Movers

**Peiyan Hu[1], Yue Wang[2]\*, Zhi-Ming Ma[1]**
[1]Academy of Mathematics and Systems Science, Chinese Academy of Sciences,
[2]Microsoft Research AI4Science
`hupeiyan18@mails.ucas.ac.cn, yuwang5@microsoft.com,`
`mazm@amt.ac.cn`

## Abstract

Recently, neural networks have been extensively employed to solve partial differential equations (PDEs) in physical system modeling. While major studies focus on learning system evolution on predefined static mesh discretizations, some methods utilize reinforcement learning or supervised learning techniques to create adaptive and dynamic meshes, due to the dynamic nature of these systems. However, these approaches face two primary challenges: (1) the need for expensive optimal mesh data, and (2) the change of the solution space's degree of freedom and topology during mesh refinement. To address these challenges, this paper proposes a neural PDE solver with a neural mesh adapter. To begin with, we introduce a novel data-free neural mesh adaptor, called Data-free Mesh Mover (DMM), with two main innovations. Firstly, it is an operator that maps the solution to adaptive meshes and is trained using the Monge-Ampère equation without optimal mesh data. Secondly, it dynamically changes the mesh by moving existing nodes rather than adding or deleting nodes and edges. Theoretical analysis shows that meshes generated by DMM have the lowest interpolation error bound. Based on DMM, to efficiently and accurately model dynamic systems, we develop a moving mesh based neural PDE solver (MM-PDE) that embeds the moving mesh with a two-branch architecture and a learnable interpolation framework to preserve information within the data. Empirical experiments demonstrate that our method generates suitable meshes and considerably enhances accuracy when modeling widely considered PDE systems. The code can be found at: `https://github.com/Peiyannn/MM-PDE.git`.

## 1 Introduction

The simulation of physical phenomena is a popular research topic in many disciplines, ranging from weather forecasting (Schalkwijk et al., 2015), structural mechanics (Panthi et al., 2007) to turbulence modeling (Garnier et al., 2021). Meanwhile, due to the rapid development of deep learning techniques, there are emerging neural network based approaches designed for simulating physical systems (Li et al., 2020; Raissi et al., 2019; Brandstetter et al., 2022; Hu et al., 2022; Gong et al., 2023). Because of the complexity of such systems and the requirement of both high accuracy and low cost, learning of physical phenomena using neural networks is still challenging.

For physical systems, traditional solvers and most neural solvers need to represent the systems' states discretely on meshes. Since the system is often non-uniform, dynamic and multi-scale, a uniform and static mesh will waste resources and lose accuracy. On the contrary, an adaptive and dynamic mesh can help allocate attention to different regions efficiently. There have been two methods (Uthman & Askes, 2005) including *h-adaption* that generates adaptive meshes by refining and coarsening cells, and *r-adaptation* or *moving mesh method* that relocates the positions of nodes. To accelerate the mesh adaptation process, deep learning techniques have been introduced to these two categories of methods (Wu et al., 2023; Yang et al., 2023; Song et al., 2022).

---

*Corresponding author.

Nonetheless, several challenges exist for the existing approaches. The majority of traditional methods are often time-consuming, particularly for dynamic systems with time-varying optimal meshes. Deep learning based approaches, such as reinforcement learning for learning $h$-adaptive meshes, exhibit low efficiency and pose difficulties in training (Wu et al., 2023; Yang et al., 2023). Moreover, $h$-adaptation does not strictly maintain a fixed number of nodes, leading to alterations in data structure and topology, complicated load-balancing, and abrupt changes in resolution (McRae et al., 2018). Another deep learning method, based on supervised learning for $r$-adaptive meshes, relies on optimal mesh data generated by traditional numerical solvers and does not consider benefiting neural networks from moving meshes, which limits its applicability (Song et al., 2022).

Targeted at these challenges in simulating dynamic systems with adaptive meshes, we propose a moving mesh based neural PDE solver (MM-PDE) whose kernel is a novel data-free neural mesh adaptor, named Data-free Mesh Mover (DMM). For the DMM, specifically, (1) to keep the degree of freedom of the numerical solution's space and avoid sharp resolution changes to promote stability, we let DMM generate $r$-adaptive meshes (moving meshes) so that the number of nodes and mesh topology are fixed. (2) To train DMM without supervised optimal mesh data accurately, we design a physical loss and a corresponding sampling strategy motivated by the concept of monitor function and optimal-transport based Monge-Ampère method. Theoretical analysis of the interpolation error bound helps us deduce the optimal form of monitor function which is crucial in the physical loss.

After obtaining the DMM, we propose MM-PDE, a graph neural network based PDE solver that relocates nodes according to the DMM. The goal is to learn the evolution while efficiently utilizing information in the data represented using the original mesh resolution. To achieve this, efficient message passing is needed both between two time steps and from the original mesh to the moving mesh. For message passing between time steps, we leverage the DMM for efficient node allocation. For message passing from the original to the moving mesh, we design a novel architecture with two key components. First, a learnable interpolation framework that generates self-adapting interpolation weights and incorporates a residual cut network. Second, the model consists of two parallel branches that pass messages on the original and moving meshes respectively, which improves performance over original mesh based GNNs.

Our main contributions are threefold: (i) We propose a neural mesh adapter DMM, which generates adaptive moving meshes and is trained in an unsupervised manner. Theoretical analysis indicates that resulting meshes minimize interpolation error. (ii) We design an end-to-end neural solver MM-PDE that adopts moving meshes from DMM to improve performance. To the best of our knowledge, it is the first moving mesh based neural solver. (iii) We conduct extensive experiments and ablation studies. Results provide strong evidence for the effectiveness and necessity of our approach.

## 2 RELATED WORKS

**Neural solvers** There are emerging deep-learning based numerical solvers in recent years. There are two main categories of methods: the physics-informed method and the data-driven method. The first one is trained with the constraint of physical loss to force the output to minimize residual of PDEs (Raissi et al., 2019) or satisfy the variational principle (Yu et al., 2018), which means that the specific form of PDEs should be known. Conversely, the second type only requires data to learn the models. These neural operators approximate mappings between spaces of functions using neural networks, such as FNO (Li et al., 2020), MP-PDE (Brandstetter et al., 2022) and DeepONet (Lu et al., 2021). However, these approaches do not take into account the meshes that serve as both carriers of data and the foundation for learning evolution.

**Traditional mesh adaptation techniques** The two most commonly deployed traditional adaptation techniques are $h$-adaptation and $r$-adaptation. $h$-adaption is also called adaptive mesh refinement, which means that mesh nodes are adaptively added or deleted (Belytschko & Tabbara, 1993; Benito et al., 2003). They suffer from certain implementation issues, such as the need for allowance for additional mesh points, and the constantly changing sparsity structure of various matrices used in calculations, which is especially essential in large computations. $r$-adaptation, or mesh movement, includes methods that move the mesh nodes and can avoid issues mentioned before. Some researchers use variational approaches to move the mesh points by solving an associated coupled system of PDEs (Winslow, 1966; Ceniceros & Hou, 2001; Tang & Tang, 2003), while in recent years, methods based on optimal transport have gained increasing attention (Budd & Williams,

2006; Weller et al., 2016; Clare et al., 2022). The primary issue with traditional methods is the time-consuming process of generating new meshes for each updated state.

**AI & Mesh adaptation** Considering the traditional mesh adaption methods are quite consuming in time and computation, deep learning techniques have been combined with mesh adaptation in many ways. Many works employ graph neural networks to learn solutions on generated meshes. For example, MeshGraphNets adopt the sizing field methodology and a traditional algorithm MINIDISK to generate refined meshes (Pfaff et al., 2020). And LAMP takes the reinforcement algorithm to learn the policy of $h$-adaption, in which case meshes' topology changes unstably (Wu et al., 2023). In addition, some researchers just focus on the generation of mesh using supervised learning (Song et al., 2022). They introduce the neural spline model and the graph attention network into their models. For improvement in both the generation and utilization of meshes, there exists considerable room.

## 3    PRELIMINARY

In this section, we introduce the definition of moving mesh. A mesh $\mathscr{T}$ can be defined as the set of all its cells $K$. To enhance solvers' accuracy and efficiency, it is natural to adjust the resolution of the mesh in different regions. For moving mesh adaptation, this adjustment is repositioning each grid point of uniform mesh to a new location. Thus we can describe the moving mesh $\mathscr{T}$ as a coordinate transformation mapping $f : [0, T] \times \Omega \to \Omega$ where $T$ is the terminal time and $\Omega$ is the domain. To learn a specific moving mesh, we need to approximate the corresponding mapping $f$.

We then introduce *monitor function* which controls the density of mesh. The monitor function $M = M(x)$ is a matrix-valued metric function on $\Omega$, and $\rho = \sqrt{\det(M(x))}$ is its corresponding *mesh density function*. The adaptive mesh moved according to $M$ is termed *M-uniform* mesh. A moving mesh $\mathscr{T}$ or coordinate transformation mapping $f$ is $M$-uniform if and only if the integrals of $\rho$ over all its cells are equal. This *equidistribution principle* can be mathematically described as

$$\int_K \rho(f(x))df(x) = \frac{\sigma}{N}, \qquad \forall K \in \mathscr{T}, \tag{1}$$

where $N$ is the number of the cells of $\mathscr{T}$ and $\sigma = \int_\Omega \rho(x)dx$ is a constant. Intuitively, the $M$-uniform mesh has larger mesh density in the region with larger value of $M$.

However, for spaces with more than one dimension, the solution of Eq (1) is not unique, which means that there is more than one way to move the mesh nodes according to $M$. To narrow it down, we seek the map closest to the identity that satisfy Eq (1). From optimal transport theories (Huang & Russell, 2010), the constraint problem is well posed, has a unique convex solution, and can be transformed into a *Monge-Ampère (MA) equation:* [1] $\det(H(\phi)) = \frac{\sigma}{|\Omega|\rho(\nabla_x\phi)}$, where $\phi$ is the potential function of the coordinate transformation function $f$, *i.e.*, $f(x) = \nabla_x\phi$, and $H(\phi) = \nabla_x^2\phi$ is the Hessian matrix of $\phi$. The equation's boundary condition is $\nabla_x\phi(\partial\Omega) = \partial\Omega$, which means the coordinate transformation maps each boundary to itself. Taking the 2-D case as an example, we can transform the boundary condition into

$$\begin{cases} \phi_\xi(0, \eta) = 0, & \phi_\xi(1, \eta) = 1, \\ \phi_\eta(\xi, 0) = 0, & \phi_\eta(\xi, 1) = 1. \end{cases}$$

To sum up, the target of seeking the $M$-uniform moving mesh is transformed to solving MA equation under boundary condition (Huang & Russell, 2010; Budd et al., 2013). To obtain the optimal way to move mesh nodes, namely the optimal monitor function, we need to analyze which $M$-uniform moving mesh corresponds to the lowest interpolation error, which is discussed in the next section.

## 4    CHOOSE MONITOR FUNCTION WITH THEORETIC GUARANTEE

In this section, we will introduce the main theorem about how to decide the optimal monitor function corresponding to the lowest interpolation error. The complete proof can be found in Appendix F.

---

[1]Monge-Ampère equation is a nonlinear second-order PDE of the form $\det D^2 u = f(x, u, Du)$.

Suppose the interpolated function $u$ belongs to the Sobolev space $W^{l,p}(\Omega) = \{f \in L^p(\Omega) : \forall |\alpha| \leq l, \partial_x^\alpha f \in L^p(\Omega)\}$, where $\alpha = (\alpha_1, \ldots, \alpha_d), |\alpha| = \alpha_1 + \ldots + \alpha_d$, and derivatives $\partial_x^\alpha f = \partial_{x_1}^{\alpha_1} \cdots \partial_{x_d}^{\alpha_d} f$ are taken in a weak sense. The error norm $e$ belongs to $W^{m,q}(\Omega)$, and $P_k(\Omega)$ is the space of polynomials of degree less than $k+1$ on $\Omega$. According to traditional mathematical theories, the form of optimal monitor function corresponding to the lowest interpolation error bound is shown in the following proposition. Note, we consider the interpolation error in the Sobolev space rather than a specific state $u$, so we deduce the interpolation error bound instead of a specific interpolation error value which is a commonly considered setting (Huang & Russell, 2010; McRae et al., 2018; Hetmaniuk & Knupp, 2007).

**Proposition 1**(Huang & Russell, 2010) On mesh $\mathscr{T}$'s every cell $K$, the optimal monitor function corresponding to the lowest error bound of interpolation from $W^{l,p}(\Omega)$ to $P_k(\Omega)$ is of the form

$$M_K \equiv \left(1 + \alpha^{-1} \langle u \rangle_{W^{l,p}(K)}\right)^{\frac{2q}{d+q(l-m)}} I, \quad \forall K \in \mathscr{T}, \tag{2}$$

where $|\cdot|_{W^{m,p}(K)} = \left(\sum_{|\alpha|=m} \int_K |D^\alpha u|^p dx\right)^{\frac{1}{p}}$ is the semi-norm of the Sobolev space $W^{m,p}(K)$, and $\langle \cdot \rangle_{W^{m,p}(K)} \equiv (1/|K|)^{1/p} \cdot |_{W^{m,p}(K)}$ is the scaled semi-norm.

From Eq (2), the optimal monitor function is proportional to $\langle u \rangle_{W^{l,p}(K)}$, so regions where derivatives are larger will have larger $M$ and thus higher mesh densities. In other words, the mesh density is positively related to the level of activity. And in Eq (2), there is a key intensity parameter $\alpha$ that controls mesh movement's intensity. When $\alpha$ increases, the mesh becomes more uniform. To decide $\alpha$, since $\rho$ is numerically calculated in practice and the error of approximating $\rho$ is non-negligible in this case, we consider this error to analyze the interpolation error bound of $M_K$ with different $\alpha$.

Taking this error into account, we then consider the reasonable choices of $\alpha$ motivated by traditional methods (Huang & Russell, 2010). Our first analysis is to take $\alpha \to 0$, in which case $M$ is dominated by $\langle u \rangle_{W^{l,p}(K)}$. For the second, $\alpha$ is taken such that $\sigma$ is bounded by a constant while $M$ is invariant under a scaling transformation of $u$.

**Theorem 1** Suppose the error of approximating $\rho$ satisfies $||\rho - \tilde{\rho}||_{L_{\frac{d}{d+q(l-m)}}(\Omega)} < \epsilon$. Also suppose $u \in W^{l+1,p}(\boldsymbol{\Omega})$, $q \leq p$ and the monitor function is chosen as in Proposition 1. Let $\Pi_k$ be the interpolation operator from $W^{l,p}(K)$ to $P_k(K)$. For notational simplicity, we let

$$B(K) = \left(\sum_K |K| \langle u \rangle_{W^{l,p}(K)}^{\frac{dq}{d+q(l-m)}}\right)^{\frac{d+q(l-m)}{dq}}.$$

(a) Choose $\alpha$ such that $\alpha \to 0$, if there exists a constant $\beta$ such that $\|D^l u(\mathbf{x})\|_{l^p} \geq \beta > 0$ a.e. in $\Omega$ holds to ensure that $M$ is positive definite, then $|u - \Pi_k u|_{W^{m,q}(\Omega)} \leq CN^{-\frac{(l-m)}{d}} B(K)$. (b) Choose $\alpha \equiv |\Omega|^{-\frac{d+q(l-m)}{dq}} B(K)$, then $|u - \Pi_k u|_{W^{m,q}(\Omega)} \leq CN^{-\frac{(l-m)}{d}} B(K)(1 + \epsilon^{\frac{d+q(l-m)}{dq}})$. In both cases, as the number of cells $N$ increases to infinity, the right-hand side of the inequality has the bound only relative to the semi-norm of $u$ as $\lim_{N \to \infty} B(K) \leq C|u|_{W^{l,\frac{dq}{d+q(l-m)}}(\Omega)}$. From the theorem, we can obtain that in case (a), the error bound is not related to $\epsilon$, while $\epsilon$ increases the interpolation error in case (b). Combining the actual condition, because the error of approximating $\rho$ is not ignorable when the mesh is not dense enough, we choose small $\alpha$ to approach the first case. So the ultimate form of monitor function utilized is Eq (2) with small $\alpha$.

## 5 A MOVING MESH BASED NEURAL SOLVER

After theoretic analysis of optimal moving meshes, in this section, we will introduce the model architecture, physics loss and sampling strategy of Data-Free Mesh Mover (DMM), innovative architecture of MM-PDE to maintain original data's information, and the effective training framework.

### 5.1 DATA-FREE MESH MOVER (DMM)

To approximate the coordinate transformation mapping $f : \mathscr{U} \times [0, T] \times \Omega \to \Omega$, we take neural operators with a physics loss and a special sampling strategy.

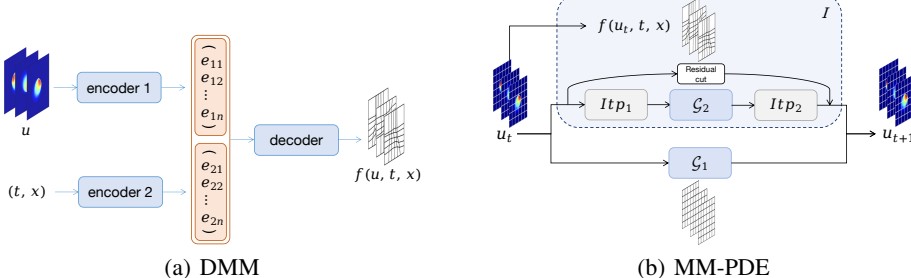

(a) DMM                                                          (b) MM-PDE

Figure 1: Model Architecture Summary: (a) Encodes state $u$ and coordinates $(t, x)$ using two networks, followed by a third network that processes their combined outputs. (b) MM-PDE has two branches: the bottom branch is a graph neural network for original meshes, and the top branch uses DMM-generated moving meshes for evolution learning. Interpolation of the discrete state onto moving meshes is done with $Itp_1$, processed by a graph neural network, then re-interpolated to original meshes using $Itp_2$. Both interpolations involve weights from $Itp_1$ and $Itp_2$. A residual cut network ensures the preservation of previous state information during interpolation.

**Model architecture** As the monitor function is related to the state $u$, it is efficient and suitable to adopt a neural operator that can generate moving meshes corresponding to different $u$. We follow and adjust the method proposed in DeepONet (Lu et al., 2021) as shown in Figure 1(a). We divide the input into two parts: the state $u$ and coordinate $(t, x)$. The former can be treated as a matrix (regular mesh) or graph (irregular mesh, *e.g.* triangular mesh), thus we encode it with a CNN and a GNN respectively. For the coordinate, we encode it with an MLP. Different from adding the product of each dimension of two outputs, since addition and first-order product are linear and do not have enough expressiveness, we concatenate two outputs and take another MLP to decode it.

**Physics loss** The next thing is to design the physics loss for the data-free training. According to Section 3, the coordinate transformation mapping $f$ is the derivative of the Monge-Ampère equation's solution. Intuitively, since mesh nodes are relocated on the basis of the uniform mesh, learning nodes' movement relative to the uniform mesh is an easier task. Consequently, we let the output of the model approximate the residual $\psi$, which can be described as

$$\nabla_x \psi = \nabla_x \phi - x = f(x) - x. \tag{3}$$

Thus the Monge-Ampère equation Eq and the boundary condition turn to

$$\det\left(H(\psi) + I_0\right) = \frac{\sigma}{|\Omega|\,\rho\left(\nabla_x\psi + x\right)}, \tag{4}$$

$$\nabla_x \psi(\partial\Omega) = 0, \tag{5}$$

where $I_0$ is the identity matrix. To learn $\psi$, we design the physics loss as the weighted sum of the loss of Monge-Ampère equation Eq (4), the loss of the boundary condition Eq (5) and the loss of $\phi$'s convex condition which will be discussed below: $l = l_{equation} + \beta l_{bound} + \gamma l_{convex}$. Here $l_{equation} = \|\,|\Omega|\,\rho(\nabla_x\psi + x)det(H(\psi) + I) - \sigma\|, l_{bound} = \|\nabla_x\psi(\partial\Omega)\|$, where $\beta$ and $\gamma$ are weights to adjust the scale of different losses. For $l_{convex}$, given that $\phi$ is convex if and only if its Hessian is positive definite, it suffices to constrain the eigenvalues of $\phi$'s Hessian matrix to be nonnegative. In the 2-D case, eigenvalues of Hessian matrix $H(\phi) = (h_{ij})_{2\times2}$ are

$$\lambda = \frac{h_{11} + h_{22} \pm \sqrt{(h_{11} + h_{22})^2 - \det H}}{2}.$$

Because the rightside of MA equation is nonnegative, the constraint can be transformed to $h_{11} + h_{22} \geq 0$, which is the same as $l_{convex} = \min\{0, \partial^2_{x_1}\phi\}^2 + \min\{0, \partial^2_{x_2}\phi\}^2$. According to the definition of $\psi$ in Eq (3), $H(\phi) = I_0 + H(\psi)$. So the final form of $l_{convex}$ is $l_{convex} = \min\{0, 1 + \partial^2_{x_1}\psi\}^2 + \min\{0, 1 + \partial^2_{x_2}\psi\}^2$.

When calculating these losses, only the mesh density function $\rho$ is calculated using the finite difference method since the input $u$ is discrete, and all other derivatives are obtained using automatic gradient computation in Pytorch (Paszke et al., 2017).

**Sampling** Because an unsuitable sampling strategy will cause difficulty in optimization and problems of getting stuck at trivial solutions (Daw et al., 2023), we design a sampling strategy specific to the Monge-Ampère equation Eq (4). Since the coordinate transformation goes dramatic with the increase of monitor function, the optimal mesh in regions of high monitor function is harder to learn and thus requires more attention. To achieve this, we let the probability of sampling points be proportional to the monitor function value, which makes the density of sampling points positively associated with the monitor function.

## 5.2  MM-PDE

After implementing DMM, We introduce MM-PDE, as shown in Figure 1(b). We focus on efficient message passing which is critical for neural PDE solvers. Efficiency is boosted in two ways: improving message passing across time steps on a discrete graph with DMM, and employing a dual-branch, three-sub-network architecture for transferring messages between original and moving meshes. This maintains data integrity in original meshes while integrating moving meshes.

Both two parallel branches serve to convey messages between time steps. The distinction lies in the fact that the message is transmitted through the original mesh and the moving mesh, respectively. This approach ensures that data within original meshes are preserved while the moving meshes are embedded. Consequently, the output of MM-PDE can be characterized as:

$$\mathscr{M}(u_t) = \mathscr{G}_1(u_t) + I(\mathscr{G}_2, \tilde{f}, u_t), \tag{6}$$

where $\mathscr{G}_1$ and $\mathscr{G}_2$ are two graph neural networks, $\tilde{f}$ is DMM, $u_t$ is the discrete representation of the state at time $t$ on the original uniform mesh, $I$ is the interpolation framework. The process of the second term is first mapping states on uniform meshes to moving meshes generated by $\tilde{f}$, then evolving the system on moving meshes to the next timestep with GNN, and finally mapping states back to uniform meshes. Every node is set to pass messages to its nearest $k$ neighbors. Since meshes are moved according to the monitor function positively related to derivatives, information passing in dynamic regions will be promoted.

The interpolation network $I$ consists of three networks: two, $Itp_1$ and $Itp_2$, for adaptive interpolation weight generation for transitions between mesh types, and a residual cut network for previous state information transfer. $Itp_1$ and $Itp_2$ generate weights for the nearest $k$ neighbors during these interpolations:$Itp_i(n_{i1}, \cdots, n_{ik}, x) = (w_{i1}, \cdots, w_{ik}), \quad i = 1, 2$, where $n_{i1}, \cdots, n_{ik}$ are coordinates of $x$'s nearest $k$ neighbors. The output of interpolation with $Itp_i$ on point $x$ is then the weighted sum $\sum_{j=1}^{k} w_{ij} u(n_{ij})$. Besides, to remedy the loss of information, we introduce a residual cut network to carry the information of previous states, whose output is added to the interpolation result of $Itp_2$.

## 5.3  TRAINING

The entire training contains two stages: training of DMM and MM-PDE. For the DMM, because of the complexity of solving a series of second order PDEs with the Hessian matrix's determinant and the requirement of accuracy for downstream tasks, we specially design the optimization algorithm. We also notice the high memory consumption and alleviate it. For the second stage, we introduce the pretraining of the interpolation framework to enhance the efficiency of message passing.

In the first stage, to enhance the optimization, after optimization with Adam optimizer, we take a traditional and more accurate optimizer BFGS. However, a key challenge we encounter during optimization is the issue of memory consumption. This is due to the neural operator requiring training on numerous collocation points across different states, with the entire state also being fed into the model. To address this issue, firstly, we input a batch of points alongside each state. This approach effectively reduces the number of inputs for every state, consequently lowering the total memory consumption. Secondly, we use BFGS optimizer to only optimize the last layer of the decoder, which also achieves great performance. Note, the memory of the A100 we use is insufficient when considering parameters of the last two layers simultaneously.

Next, MM-PDE is trained while parameters of DMM are fixed. To further enhance the accuracy of interpolation, the interpolation network framework $I$ is first pretrained on states in the training set.

Table 1: Results of Burgers' equation and flow around a cylinder.

(a) Meshes from DMM on the Burgers' equation.

| METRIC | ORIGINAL UNIFORM MESH | MESH FROM DMM |
|---|---|---|
| std | 0.1027 | **0.0469** |
| range | 0.7524 | **0.2061** |

(b) MM-PDE and baselines on the Burgers' equation.

| MODEL | ERROR | TIME(s) |
|---|---|---|
| MM-PDE | **1.04e-05** | 0.5192 |
| GNN | 3.10e-05 | 0.3078 |
| bigger GNN | 1.19e-05 | 0.3298 |
| CNN | 1.26e-05 | 0.0027 |
| FNO | 1.18e-05 | 0.0159 |
| LAMP | 1.61e-05 | 1.4598 |

(c) Meshes from DMM on flow around a cylinder.

| METRIC | ORIGINAL UNIFORM MESH | MESH FROM DMM |
|---|---|---|
| std | 0.0136 | **0.0094** |
| range | 0.1157 | **0.0733** |

(d) MM-PDE and baselines on flow around a cylinder.

| MODEL | ERROR |
|---|---|
| MM-PDE | **0.0846** |
| GNN | 0.2892 |
| bigger GNN | 0.1548 |
| CNN | - |
| FNO | - |
| LAMP | 0.4040 |
| Geo-FNO | 0.2844 |

In this process, we map every state on the original mesh to its corresponding moving mesh and then back to the original without evolution to the next state. The pretraining loss can be described as

$$l_{pre} = MSE(I(\mathscr{G}_I, \tilde{f}, u), u),$$

where $\mathscr{G}_I$ is the fixed identity mapping, $MSE$ is the mean squared error. Then the training loss is the same as common autogressive methods, which is the error between predicted states and groundtruth.

## 6 EXPERIMENTS

In the experiments, we evaluate our method on two datasets. The first is a time dependent Burgers' equation, and the second is the flow around a circular cylinder. Our target is to verify that our method can generate and utilize moving meshes reasonably, and is superior in accuracy. Also, to further demonstrate the effectiveness of components in MM-PDE and discuss other possible training patterns of the mesh mover, we do ablation studies to evaluate other variants of MM-PDE.

**Baselines** We compare MM-PDE with GNN (Brandstetter et al., 2022), bigger GNN, CNN, FNO (Li et al., 2020) and LAMP (Wu et al., 2023) under mean squared error, where GNN is the same as the GNN component in MM-PDE, and bigger GNN is deepened and widened to have a close number of parameters to MM-PDE. As LAMP adjusts vertexes' number dynamically, we keep it almost fixed to make the computation cost the same as others.

**Evaluation metric** To provide quantitative results, we design some evaluation metrics. For DMM, as shown in Eq (1), the optimal moving mesh is the one satisfying the equidistribution principle under optimal monitor function $M$, which means mesh movement uniformizes cells' volumes under $M$. Thus we calculate the standard deviation (std) and range of all cells' volumes. As for MM-PDE, we choose MSE referring to previous works (Brandstetter et al., 2022; Wu et al., 2023).

### 6.1 BURGERS' EQUATION

We first consider the 2-D Burgers' equation, which has wide applications in fields related to fluids.

$$\frac{\partial u}{\partial t} + (u \cdot \nabla)u - \nu\nabla^2 u = 0, \quad (t, x_1) \in [0, T] \times \Omega.$$

The DMM has a parameter count of 1222738, with a training time of 7.52 hours and memory usage of 15766MB. The results are reported in Table 1(a). Compared with original uniform meshes, meshes from DMM have a much lower standard deviation and range, which means the volumes of

cells under $M$ are more even. So DMM moves meshes to satisfy the equidistribution condition, which is equivalent to the optimal moving mesh's definition.

From Table 1(b), our method shows the lowest error and has comparable inference time to the base model MP-PDE, though longer than CNN and FNO due to its architecture. Compared to GNN, moving meshes notably boost performance. Increasing GNN's depth and width only marginally improves it, indicating our performance gains aren't just from more parameters. Our moving mesh method and physics-informed training outperform LAMP's mesh refinement and RL.

## 6.2 FLOW AROUND A CYLINDER

The next case is the simulation of a real physical system, and has more complicated dynamics than the first experiment. We test the models on the flow around a circular cylinder.

The DMM has 2089938 parameters, with a training time of 22.91 hours and memory usage of 73088MB. Due to the irregular shape of the original mesh, FNO and CNN are not applicable. We thus compare with Geo-FNO (Li et al., 2022). As shown in Table 1(c), because the std and range of cells' volumes after DMM's movement are greatly lower than the original, our proposed method for moving meshes can significantly uniformize the volumes of cells under metric $M$. The visualizations of moving meshes and rollout results of MM-PDE are shown in Appendix E.1 and E.2.

From Table 1(d) and 11, MM-PDE has the best performance in simulation of a real and complex physical system as well. Compared with GNN and bigger GNN, the embedding of DMM considerably improves accuracy and this improvement is not brought by increasing the parameters' number. Also, the mesh mover trained with the Monge-Ampère equation shows progress over LAMP's mesh refinement by reinforcement learning. Relative MSE results are reported in Appendix E.1.

## 6.3 3-D GS EQUATION

Finally, we extend DMM and MM-PDE to 3-D scenarios to demonstrate the universality of our method. We consider the 3-D Gray-Scott (GS) reaction-diffusion equation with periodic boundaries The results are reported in Appendix B, suggesting MM-PDE's scalability.

## 6.4 ABLATION STUDY

To further discuss advantages of our proposed method, we provide results of 8 ablation studies.

**Physics loss optimization** Table 2 reports physics loss of DMM on the Burgers' equation and flow around a cylinder. From the tables, physics loss decreases substantially, and is lower after BFGS optimizer's optimization.

Table 2: Physics loss for DMM before and after optimization using the Adam and BFGS optimizers

(a) Burgers' equation.

| LOSS | INIT | ADAM | BFGS |
|------|------|------|------|
| $l$ | 28.248 | 0.045 | 0.026 |
| $l_{equation}$ | 28.209 | 0.044 | 0.023 |
| $l_{bound}$ | 3.94e-4 | 1.07e-06 | 2.80e-06 |
| $l_{convex}$ | 2.47e-7 | 0.0 | 0.0 |

(b) Flow around a cylinder.

| LOSS | INIT | ADAM | BFGS |
|------|------|------|------|
| $l$ | 0.881 | 0.101 | 0.095 |
| $l_{equation}$ | 0.547 | 0.095 | 0.093 |
| $l_{bound}$ | 3.34e-4 | 5.99e-6 | 1.54e-6 |
| $l_{convex}$ | 0.0 | 0.0 | 0.0 |

**Resolution transformation (Appendix C.1)** A DMM's advantage is its ability to generate meshes of varying resolutions during inference. But due to the selected encoder 1's inability to handle varying resolutions, this requires interpolation prior to inputting states of different resolutions. We assess bicubic interpolation and the method in Eq 16, with training data at $48 \times 48$ and 2521 resolutions. The meshes from the pre-trained DMM are illustrated in Figure 2, and associated quantitative results are in Table 7 and 8. The data, regardless of resolution, adheres well to the MA equation.

**Mesh tangling and $l_{convex}$ (Appendix C.2)** When the Jacobian determinant of $f$ is negative, moving meshes will encounter the issue of mesh tangling, where edges become intertwined. As we use meshes solely to enhance neural solvers, this issue does not pose a problem in our context. But

we explore it for future potential expansions of DMM's applications. Constraining the Jacobian determinant of $f$ to be non-negative is equivalent to requiring that $\Phi$ is convex, which is already constrained by $l_{convex}$. The trend of $l_{convex}$ during training is illustrated in Figure 3, showing that it can reach a very low level early in the training process. Additionally, we find that the generated grid points of test data has $l_{convex} = 0$. These demonstrate that this constraint is well met.

**Effects of each component of MM-PDE (Appendix C.4)** To show the effects of each component of MM-PDE, we delete or replace some parts. We select 'no $\mathscr{G}_1$', '$\mathscr{G}_1 + \mathscr{G}_2$', 'no Residual' and 'uniform mesh' to compare with MM-PDE. As shown in Table 9, a well-defined and suitable moving mesh is crucial for maintaining accuracy. And our introduced ways to preserve information, including the branch $\mathscr{G}_1$ on original meshes and the residual cut network, are effective.

**Roles of three losses within the physics loss (Appendix C.3)** We respectively reduce weights of the three losses within the physics loss by a factor of ten for training. Table 3 displays the std and range of meshes generated by new DMMs. When the weight of $l_{equation}$ is decreased, the std and range increase the most, suggesting that the MA equation is not adequately satisfied. The result does not deviate significantly from the original when the weight of $l_{convex}$ is decreased, as $l_{convex}$ can already reach a low level as stated in Appendix C.2. Figure 4 shows the plotted meshes of decreased $l_{bound}$, implying that boundary conditions are not as well satisfied as before.

Table 3: Results of different loss weights. The three weights are respectively weights of $l_{equation}$, $l_{bound}$ and $l_{convex}$, and (1,1000,1) is the original weight.

| METRIC | ORIGINAL UNIFORM MESH | (1,1000,1) | (1,100,1) | (1,1000,0.1) | (0.1,1000,1) |
|--------|----------------------|-----------|-----------|--------------|--------------|
| std    | 0.103                | 0.047     | 0.048     | 0.047        | 0.054        |
| range  | 0.752                | 0.206     | 0.239     | 0.225        | 0.265        |

**Relationship between std, range and MM-PDE** To further explore the relationship between std and range of moving meshes and MM-PDE's performance, we introduce a weighted combination of the identity map $f_I$ and $\tilde{f}$ to replace $\tilde{f}$. Specifically, $\mathscr{M}(u_t) = \mathscr{G}_1(u_t) + I(\mathscr{G}_2, \alpha \tilde{f} + (1-\alpha)f_I, u_t)$. Table 4 shows the monotonic relation between std, range and MM-PDE.

Table 4: Relationship between std, range and error. Specifically, $f$ of MM-PDE is $\alpha \tilde{f} + (1-\alpha)f_I$.

| $\alpha$ | std    | range  | ERROR    |
|----------|--------|--------|----------|
| 0.8      | 0.0581 | 0.3432 | 6.154e-7 |
| 0.6      | 0.0708 | 0.4685 | 8.714e-7 |
| 0.4      | 0.0826 | 0.5695 | 9.209e-7 |
| 0.2      | 0.0939 | 0.6636 | 1.224e-6 |

**MM-PDE with MGN (Appendix A)** We conduct tests on the Burgers' equation using the Mesh Graph Net as the GNN component, and additionally introduced the Mesh Graph Net as a baseline (Pfaff et al., 2020). Results in Table 5 shows that our methods still works.

**Training ways of DMM (Appendix C.5)** As DMM is the core component of MM-PDE, we also explore the influence of training ways of MM-PDE's moving mesh component. We consider learning DMM only when training MM-PDE (**1. end2end**) and setting trained DMM also learnable during training of MM-PDE (**2. mm+end2end**). Results in Table 10 display that mm+end2end achieves the lowest error, implying DMM trained with the physics loss is a nice initial training state.

# 7 CONCLUSION AND DISCUSSION

In this paper, we introduce the Data-free Mesh Mover (DMM) for learning moving meshes without labeled data, choosing the monitor function $M$ through theoretical analysis. We then present MM-PDE, a neural PDE solver using DMM, with moving and original meshes for information transfer and a neural interpolation architecture. We note that the current monitor function, based on the state at time $t$, may not optimally capture the variation of future states. Future work could consider how to incorporate future state information in DMM's design and training, possibly through a surrogate model predicting the next moment.

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

## A    EXPERIMENT SETTINGS

We first consider the 2-D Burgers' equation, which has wide applications in fields related to fluids.

$$\frac{\partial u}{\partial t} + (u \cdot \nabla)u - \nu \nabla^2 u = 0, \quad (t, x_1) \in [0, T] \times \Omega,$$
$$u(t, 0, x_2) = u(t, 1, x_2), \quad u(t, x_1, 0) = u(t, x_1, 1), \quad (\text{Periodic BC}),$$
$$u(0, x_1, x_2) = u_0(x_1, x_2), \quad x \in \Omega,$$

where $u$ is the velocity field, $\nu = 0.1$ is the viscosity and $u_0 = exp(-100(x_1 - \alpha)^2 - 100((x_1 - (1 - \beta))^2 + (x_2 - (1 - \beta))^2))$ is the initial condition where $\alpha$ and $\beta$ are uniformly sampled in $[0, 1]$.

The equation is defined on $\Omega = [0, 1] \times [0, 1]$ and the length of trajectories is $T = 29s$. To ensure the accuracy of the numerical solution, we solve the equation numerically on the uniform $192 \times 192$ grid with 30 timesteps using a simulation toolkit (Holl et al., 2020), and one timestep corresponds to one second. When training and testing the models, we downsample the solution to $48 \times 48$. All the models autogressively predict the solution, taking one timestep as input.

Table 5: MM-PDE combined with the Mesh Graph Net.

| MODEL | ERROR | TIME(s) |
|---|---|---|
| MM-PDE(MGN) | 2.13e-05 | 0.1187 |
| MGN | 2.46e-05 | 0.0400 |

The next case is the simulation of a real physical system, and has more complicated dynamics than the first experiment. We test the models on the flow around a circular cylinder. More precisely, a circular cylinder of radius $r = 0.04$ centered at $(0.06, 0.25)$ is replaced in an incompressible and inviscid flow in the $[0, 0.5] \times [0, 0.5]$ square. The viscosity coefficient $\nu$ is set to be $10^{-3}$.

We use FEniCS (Scroggs et al., 2022), a finite element computing toolkit, to generate the velocity field based on the Navier-Stokes equation. The solver rolls out 40 timesteps with $dt = 0.1s$ on 2521 triangular lattices. Before $0.9s$, the flow is under a constant external force $f$ with magnitude randomly sampled in $[-20, 20]$. Then horizontal velocity after $0.9s$ are used as training and test sets.

The third case is the 3-D Gray-Scott (GS) reaction-diffusion equation with periodic boundaries and generate data using the high-order finite difference (FD) method as referenced in prior literature (Ren et al., 2023). The data are then downsampled uniformly to a $24 \times 24 \times 24$ grid with $dx = 100/24$, spanning across 11 time steps with $dt = 20$. As the solution of 3-D GS equation is in the form of $u(x, y, z) = (u_1(x, y, z), u_2(x, y, z))$, we take the first dimension of solutions as training and test data.

## B    3-D GS EQUATION

In this section, we report the results on 3-D GS reaction-diffusion equation in Table 6. From this table, it can be observed that our proposed model architecture can be extended to 3-dimensional equations. Furthermore, it outperforms the baselines, demonstrating excellent performance.

Table 6: Results of the 3-D GS equation.

| MODEL | MSE |
|---|---|
| MM-PDE | **1.852e-04** |
| GNN | 2.106e-04 |

## C  ABLATION STUDY

### C.1  RESOLUTION TRANSFORMATION

In this section, we further demonstrate the effectiveness of our method in resolution transformation. One advantage of DMM is that, after training, it can generate meshes of any resolution during the inference stage. The only change needed is that, since the encoder 1 we select to encode states cannot handle data of any resolution, when we input states of other resolutions, we first need to interpolate them. We evaluate two kinds of interpolation: bicubic interpolation for the Burgers' equation and the interpolation method in Eq 16 for the flow around a cylinder. The meshes generated by previously trained DMM are reported in Figure 2. The quantitative results obtained are shown in Table 7 and 8, where the training data of DMM is of $48 \times 48$ and 2521 resolution. It displays that data of no matter lower or higher resolution can still satisfy the MA equation well.

Table 7: Test on data of different resolutions on the Burgers' equation.

| Metric | Uniform $24 \times 24$ Mesh | $24 \times 24$ Mesh from DMM |
|---|---|---|
| std | 0.3315 | 0.1596 |
| range | 1.6879 | 0.7055 |
| Metric | Uniform $96 \times 96$ Mesh | $96 \times 96$ Mesh from DMM |
| std | 0.0304 | 0.0136 |
| range | 0.3273 | 0.0858 |

Table 8: Test on data of different resolutions on the flow around a cylinder.

| Metric | Original 2521 Mesh | 3286 Mesh from DMM |
|---|---|---|
| std | 0.0112 | 0.0077 |
| range | 0.1111 | 0.0665 |
| Metric | Original 2521 Mesh | 1938 Mesh from DMM |
| std | 0.0163 | 0.0111 |
| range | 0.1488 | 0.0833 |

### C.2  MESH TANGLING AND $l_{convex}$

Moving meshes can sometimes encounter the issue of mesh tangling, where the edges of different meshes become intertwined. This situation corresponds to cases where the Jacobian determinant of the coordinate transformation $f$ is negative. Although this issue does not pose a problem in our context as we are using the meshes generated by DMM solely to enhance the performance of neural solvers, we nevertheless explore this issue for future potential expansions of DMM's range of applications.

Constraining the Jacobian determinant of $f$ to be non-negative is equivalent to requiring that $\Phi$ is a convex function. This condition is already constrained by $l_{convex}$ in the physics loss of DMM. The trend of $l_{convex}$ during the training process is illustrated in Figure 3, showing that it can reach a very low level early in the training process. Additionally, a test is carried out on the generated grid points of test data, resulting in an $l_{convex}$ of 0. This demonstrates that this constraint is well met during the training process.

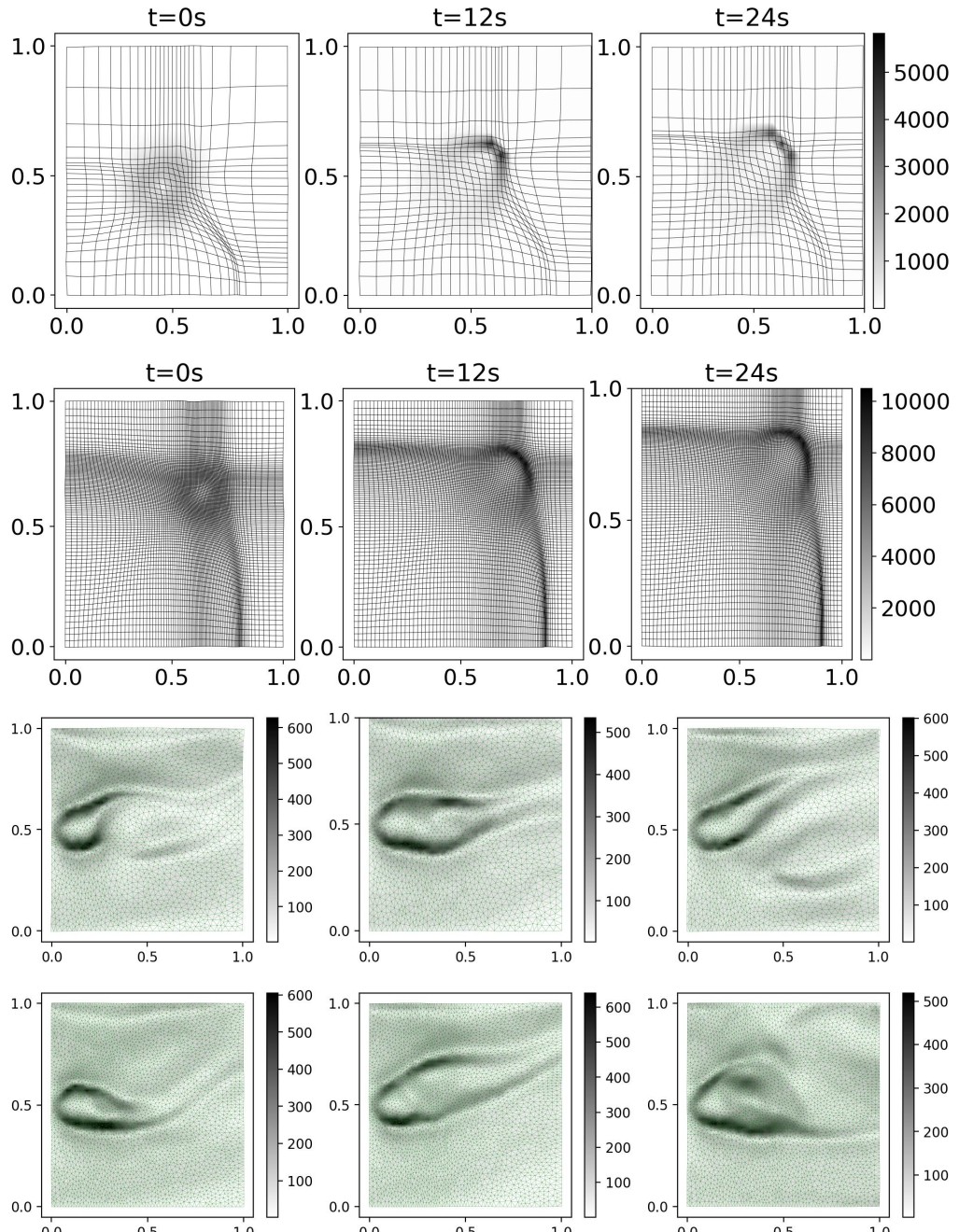

Figure 2: Meshes generated by previously trained DMM with different resolution data. Four lines respectively correspond to $24 \times 24$ resolution, $96 \times 96$ resolution, 1938 resolution and 3286 resolution.

### C.3    ROLES OF THREE LOSSES WITHIN THE PHYSICS LOSS

To investigate the roles of the three losses within the physics loss, we reduce their respective weights by a factor of ten for training. Table 3 displays the std and range of meshes generated by DMM after training, where three weights are respectively weights of $l_{bound}$, $l_{convex}$ and $l_{equation}$, and (1,1000,1) is the original weight. It can be observed that when the weight of $l_{equation}$ is decreased, the std and range increase the most. This suggests that the Monge-Ampère (MA) equation is not adequately satisfied, implying that the volume of cells under $M$ is not uniformly distributed. When

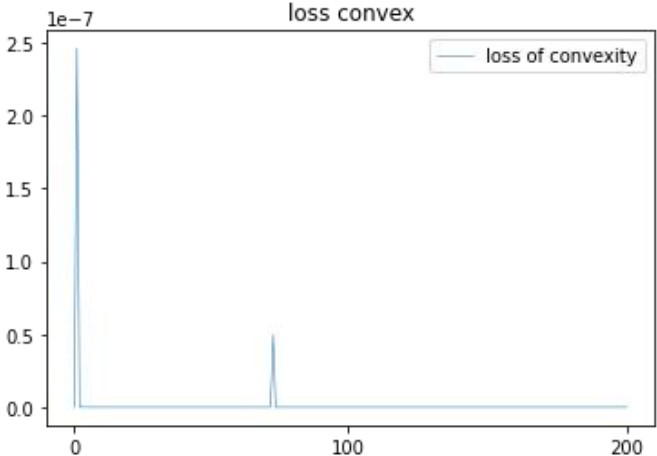

Figure 3: $l_{convex}$ of the Burgers' equation during training. The horizontal axis represents the number of training epochs, while the vertical axis depicts the value of the loss.

the weight of $l_{convex}$ is decreased, the result does not deviate significantly from the original, as $l_{convex}$ can already reach a very low level as stated in Appendix C.2. To observe the effect of $l_{bound}$, Figure 4 shows the plotted meshes. It is evident that when the weight of $l_{bound}$ is decreased, the boundary conditions are not as well satisfied as before.

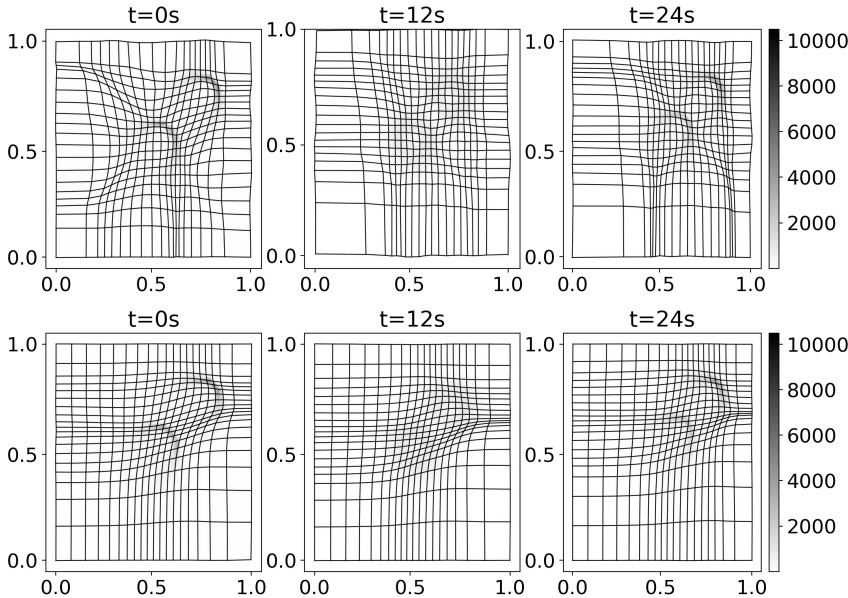

Figure 4: Moving meshes of DMM with reduced weight of $l_{bound}$ (top) and $l_{equation}$ (bottom).

## C.4 VARIANTS OF MM-PDE

We respectively test the effectiveness of different components of MM-PDE. To test necessity of the two-branch architecture, the first variant is to delete the branch $\mathscr{G}_1$ on original meshes (**1. no $\mathscr{G}_1$**), resulting in $\mathscr{M}(u_t) = I(\mathscr{G}_2, f, u_t)$. The second is to delete the residual cut network (**2. no Residual**), which aims to prove its ability to maintain information in the interpolation framework. We also replace the branch on moving meshes with another branch on original meshes without the interpolation framework (**3. $\mathscr{G}_1 + \mathscr{G}_2$**) to display the importance of moving meshes, whose output can be formulated as $\mathscr{M}(u_t) = \mathscr{G}_1(u_t) + \mathscr{G}_2(u_t)$. And to show the validity of meshes'

Table 9: Results of variants of MM-PDE.

| MODEL | ERROR |
|---|---|
| MM-PDE | **6.63e-07** |
| no $\mathscr{G}_1$ | 1.258e-06 |
| $\mathscr{G}_1 + \mathscr{G}_2$ | 8.852e-07 |
| no Residual | 8.467e-07 |
| uniform mesh | 1.324e-06 |

Table 10: Results corresponding to different training was of DMM

| MODEL | ERROR |
|---|---|
| end2end | 6.640e-07 |
| mm+end2end | **3.447e-07** |

optimality, we replace moving mesh with uniform mesh in MM-PDE (**4. uniform mesh**). In this case, $\mathscr{M}(u_t) = \mathscr{G}_1(u_t) + I(\mathscr{G}_2, f_I, u_t)$, where $f_I$ is the identity map.

As shown in Table 9, we can conclude that our introduced ways to preserve information, including the branch $\mathscr{G}_1$ on original meshes and the residual cut network, are effective. Besides, a well-defined and suitable moving mesh is crucial for maintaining accuracy. The absence of moving meshes results in a decrease in accuracy, and the use of uniform meshes can not bring enhancement.

### C.5 TRAINING WAYS OF DMM

## D EXTENDED RESULTS OF THE BURGERS' EQUATION

To let the readers see clearly, we first present moving meshes of $20 \times 20$ resolution generated by the DMM in Figure 5. And then we provide a larger image of the mesh on the $48 \times 48$ grid in Figure 6 The DMM used to generate meshes in the main text is the same as the one used here.

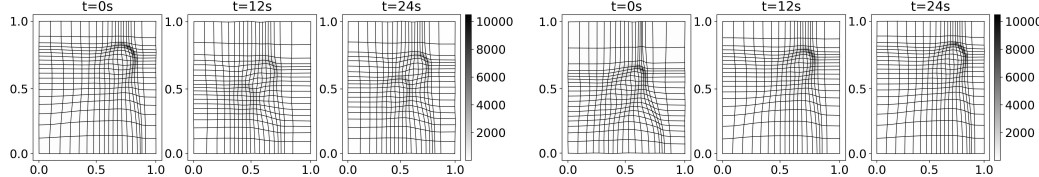

Figure 5: Moving meshes of the 2-D Burgers' equation generated by DMM. The left three figures and the right three ones present moving meshes of $20 \times 20$ resolution and monitor functions of two trajectories at $t = 0, 12, 24s$ respectively.

Furthermore, we examine the quality of meshes when the value of the monitor function is extremely large near boundaries to verify the quality of the meshes generated by DMM, as this represents a more complex scenario. Given that the Burgers' equation we select has periodic boundaries, we choose some states where the maximum value of the monitor function is relatively close to the boundary to generate meshes. Additionally, we generate new states with maximum values even closer to boundaries and obtained the new mesh. These diagrams are all displayed in Figure 7, demonstrating that DMM can also handle such situations.

## E EXTENDED RESULTS OF FLOW AROUND A CYLINDER

### E.1 VISUALIZATION OF ROLLOUT RESULTS

To more intuitively present the result of the experiment on a flow around a cylinder, we display the relative Mean Squared Error (RMSE) of models in Table 11, which shows relatively low values. Although the MSE appears large in Table 1(d), this is due to the overall large norm of the data.

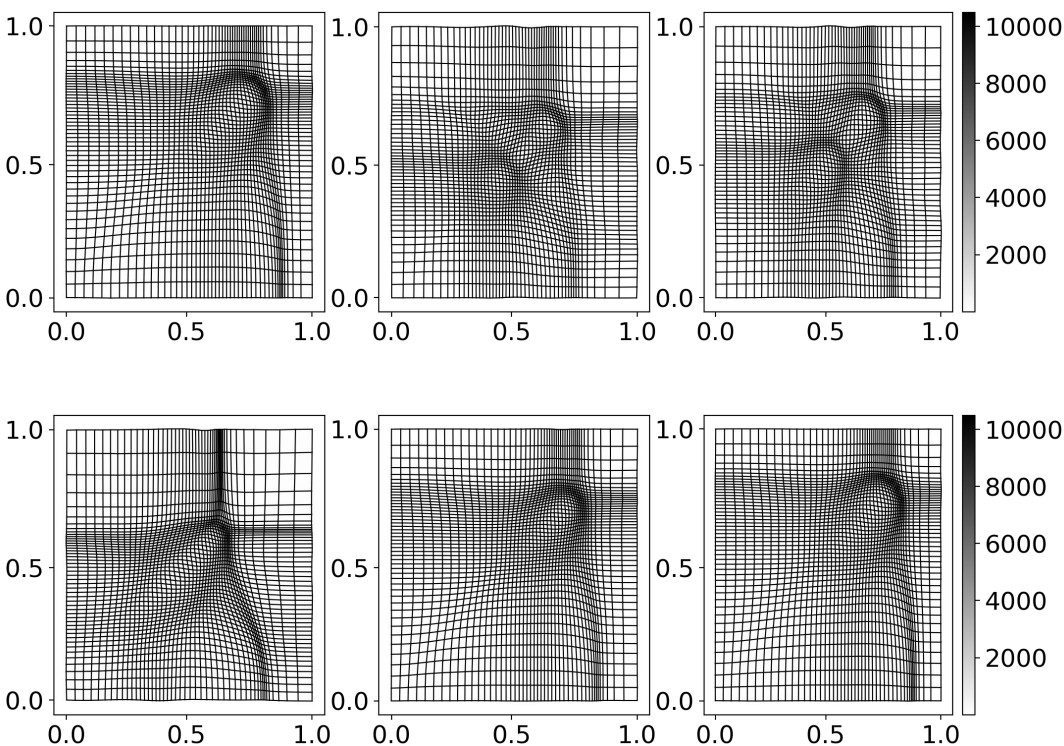

Figure 6: $48 \times 48$ moving meshes of the Burgers' equation generated by DMM. The bars on the right present the correspondence between color and the value of the monitor function.

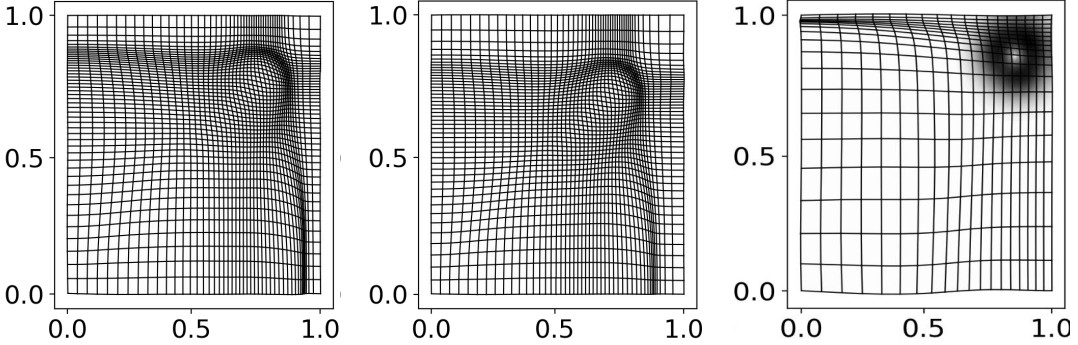

Figure 7: Large monitor function near boundaries. The two figures on the left are states from the original dataset, while the figure on the right represents a newly generated state with a large M value closer to the boundary.

Moreover, in Figure 8, we depict the real trajectory alongside the trajectory generated by MM-PDE. It can be observed that the data produced by MM-PDE aligns well with the real trajectory.

### E.2 MOVING MESH VISUALIZATION

For flow around a cylinder, we present additional visualizations of moving meshes generated by DMM in Figure 9. The pictures show that the triangular cells' density is higher in regions with higher monitor functions, which verifies the equidistribution principle intuitively.

Table 11: RMSE of flow around a cylinder.

| MODEL | RMSE |
|---|---|
| MM-PDE | **0.0197** |
| GNN | 0.0649 |
| bigger GNN | 0.0349 |
| CNN | - |
| FNO | - |
| LAMP | 0.0693 |
| Geo-FNO | 0.0602 |

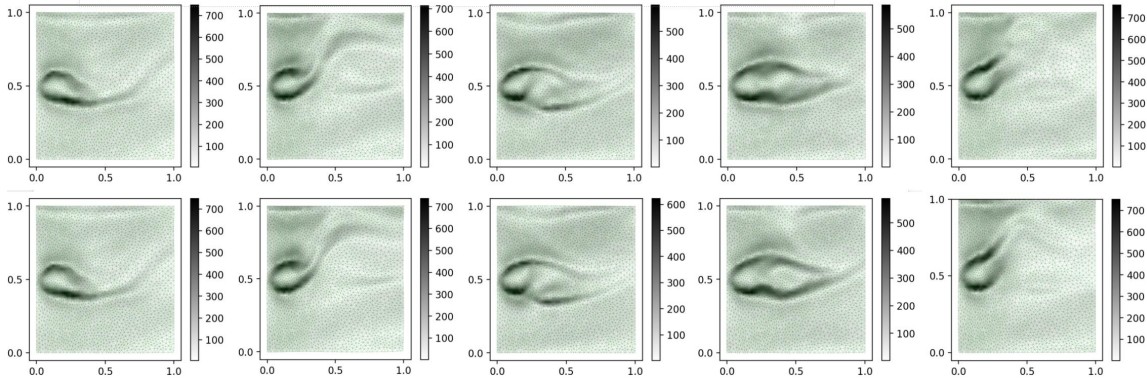

Figure 8: Ground truth (top) and trajectories generated by MM-PDE (bottom).

## F  THEORY

In this section, we provide the proof of the proposition and theorem mentioned in Section 4. We first introduce three basic lemmas for preparation of proof (Huang & Russell, 2010).

In the mathematical analysis below, every cell is defined as a couple $(K, P_K)$, where $K$ is a mesh cell, $P$ is a finite-dimensional linear space of functions defined on $K$ that are used to approximate the solution.

**Lemma 1.1** (Ciarlet, 2002) Let $(K, P_K)$ be a cell associated with the simplicial reference element $K$. For some integers $m, k,$ and $l : 0 \leq m \leq l \leq k + 1$, and some numbers $p, q \in [1, \infty]$,

$$W^{l,p}(K) \hookrightarrow W^{m,q}(K),$$
$$P_k(K) \subset P_K \subset W^{m,q}(K),$$

where $P_k(K)$ is the space of polynomials of degree $\leq k$. Then, for any $u \in W^{l,p}(K)$

$$|u - \Pi_{k,K} u|_{W^{m,q}(K)} \leq C \|(F_K')^{-1}\|^m \cdot \|F_K'\|^l \cdot |K|^{\frac{1}{q} - \frac{1}{p}} \cdot |u|_{W^{l,p}(K)}, \tag{7}$$

where $\Pi_{k,K} : W^{l,p}(K) \to P_K$ denotes the $P_K$-interpolation operator on $K$, $F_K'$ is the Jacobian matrix of the affine mapping between two simplicial cells $(K, P_K)$, and $(\hat{K}, \hat{P}_K)$, and $\| \cdot \|$ denotes the $l_2$ matrix norm.

**Lemma 1.2** For any matrix $A \in \mathbb{R}^{d \times d}$,

$$tr(A^T A) = tr(AA^T) = \|A\|_F^2,$$

where $\| \cdot \|_F$ denotes the Frobenius matrix norm.

**Lemma 1.3** An $M$-uniform mesh satisfies

$$\frac{1}{d} tr((F_K')^T M_K F_K') = \det((F_K')^T M_K F_K')^{\frac{1}{d}}, \quad \forall K \in \mathcal{T}_h,$$

$$\frac{1}{d} tr((F_K')^{-1} M_K^{-1} (F_K')^{-T}) = \det((F_K')^{-1} M_K^{-1} (F_K')^{-T})^{\frac{1}{d}}, \quad \forall K \in \mathcal{T}_h.$$

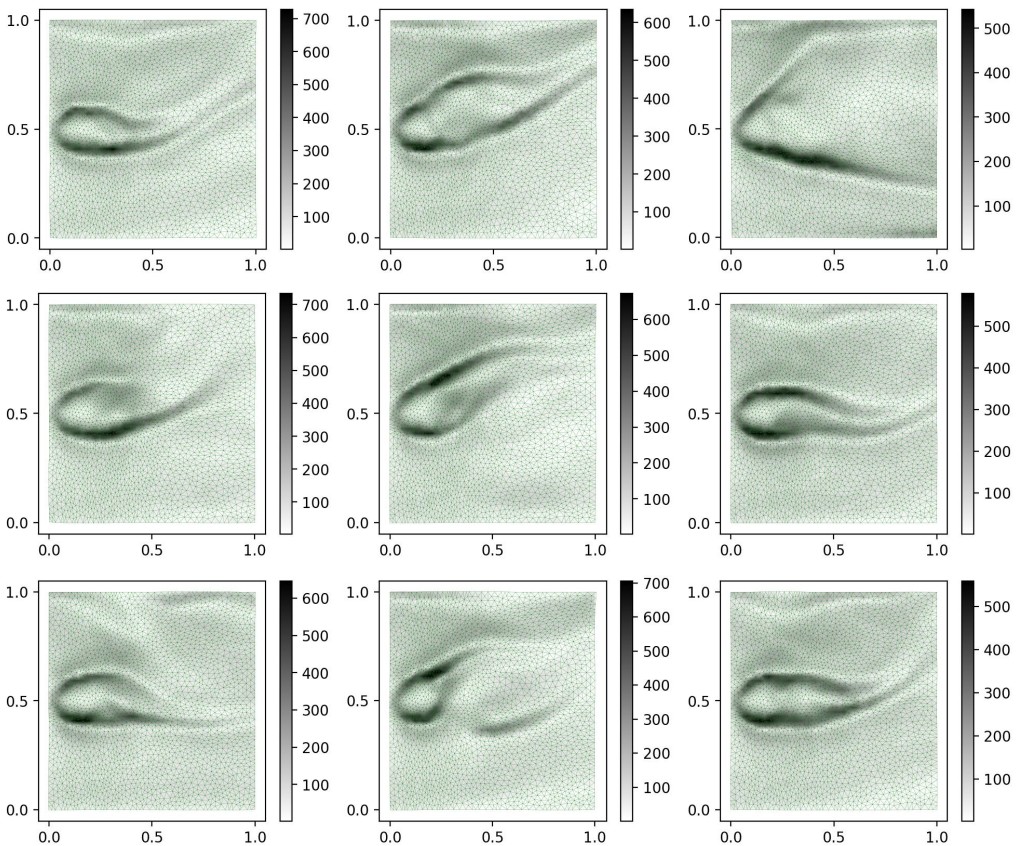

Figure 9: Moving meshes of the flow around a cylinder generated by DMM. The bars on the right present the correspondence between color and the value of the monitor function.

Next, we give the proof of the Poposition 1, which indicates the form of optimal monitor function $M$. The monitor function has a regularization parameter $\alpha$. It is often referred to as the intensity parameter since it controls the level of intensity of mesh concentration. As $\alpha$ increases, the mesh becomes more uniform.

**Proposition 1** The optimal monitor function is of the form

$$M_K \equiv \left(1 + \alpha^{-1} \langle u \rangle_{W^{l,p}(K)}\right)^{\frac{2q}{d+q(l-m)}} I, \quad \forall K \in \mathscr{T},$$

$$\rho_K \equiv \left(1 + \alpha^{-1} \langle u \rangle_{W^{l,p}(K)}\right)^{\frac{dq}{d+q(l-m)}}, \quad \forall K \in \mathscr{T}.$$

Proof: Taking the $q$-th power on both sides of Eq(7) and summing over all of the cells, we obtain, for $u \in W^{l,p}(\boldsymbol{\Omega})$

$$|u - \Pi_k u|^q_{W^{m,q}(\Omega)} \le C \sum_K |K| \cdot \|(F_K')^{-1}\|^{mq} \cdot \|F_K'\|^{lq} \cdot \langle u \rangle^q_{W^{l,p}(K)}, \tag{8}$$

where

$$\langle u \rangle_{W^{l,p}(K)} = \left(\frac{1}{|K|} \int_K \sum_{i_1,\dots,i_l} \left|D^{(i_1,\dots,i_l)} u\right|^p dx\right)^{\frac{1}{p}} = \left(\frac{1}{|K|} \int_K \|D^l u\|^p_{l_p} dx\right)^{\frac{1}{p}}.$$

Since

$$\|F_K'\| \le \|F_K'\|_F, \qquad \|(F_K')^{-1}\| \le \|(F_K')^{-1}\|_F,$$

from Lemma 1.2 we have

$$\|F'_K\|^2 \leq \operatorname{tr}\left((F'_K)^T(F'_K)\right), \quad \|(F'_K)^{-1}\|^2 \leq \operatorname{tr}\left((F'_K)^{-1}(F'_K)^{-T}\right).$$

Thus Eq(9) can be rewritten as

$$|u - \Pi_k u|^q_{W^{m,q}(\Omega)} \leq C\sum_K |K| \cdot \left[\frac{1}{d}\operatorname{tr}\left((F'_K)^{-1}(F'_K)^{-T}\right)\right]^{\frac{mq}{2}} \times \left[\frac{1}{d}\operatorname{tr}\left((F'_K)^T(F'_K)\right)\right]^{\frac{lq}{2}} \cdot \langle u\rangle^q_{W^{l,p}(K)} \cdot$$

$$\leq C\sum_K |K| \cdot \left[\frac{1}{d}\operatorname{tr}\left((F'_K)^{-1}(F'_K)^{-T}\right)\right]^{\frac{mq}{2}} \times \left[\frac{1}{d}\operatorname{tr}\left((F'_K)^T(F'_K)\right)\right]^{\frac{lq}{2}}$$
$$\cdot \left(\alpha + \langle u\rangle_{W^{l,p}(K)}\right)^q$$

$$\leq C\alpha^q\sum_K |K| \cdot \left[\frac{1}{d}\operatorname{tr}\left((F'_K)^{-1}(F'_K)^{-T}\right)\right]^{\frac{mq}{2}} \times \left[\frac{1}{d}\operatorname{tr}\left((F'_K)^T(F'_K)\right)\right]^{\frac{lq}{2}}$$
$$\cdot \left(1 + \alpha^{-1}\langle u\rangle_{W^{l,p}(K)}\right)^q.$$

Denote

$$E(\mathscr{T}) = N^{\frac{(l-m)q}{d}}\sum_K |K| \cdot \left[\frac{1}{d}\operatorname{tr}\left((F'_K)^{-1}(F'_K)^{-T}\right)\right]^{\frac{mq}{2}} \times \left[\frac{1}{d}\operatorname{tr}\left((F'_K)^T(F'_K)\right)\right]^{\frac{lq}{2}} \tag{9}$$

$$\cdot \left(1 + \alpha^{-1}\langle u\rangle_{W^{l,p}(K)}\right)^q. \tag{10}$$

Then the task for finding the optimal monitor function becomes

$$\min E(\mathscr{T}(M)).$$

From the arithmetic-mean geometric-mean inequality and $\det(F'_K) = |K|$, we have

$$E(\mathscr{T}(M)) = N^{\frac{(l-m)q}{d}}\sum_{\mathcal{K}} |K| \cdot \left[\frac{1}{d}\operatorname{tr}\left((F'_K)^{-1}(F'_K)^{-T}\right)\right]^{\frac{mq}{2}} \times \left[\frac{1}{d}\operatorname{tr}\left((F'_K)^T(F'_K)\right)\right]^{\frac{lq}{2}}$$
$$\cdot \left(1 + \alpha^{-1}\langle u\rangle_{W^{l,p}(K)}\right)^q$$

$$\geq N^{\frac{(l-m)q}{d}}\sum_K |K| \cdot \left[\det\left((F'_K)^{-1}(F'_K)^{-T}\right)\right]^{\frac{mq}{2d}} \times \left[\det\left((F'_K)^T(F'_K)\right)\right]^{\frac{lq}{2d}}$$
$$\cdot \left(1 + \alpha^{-1}\langle u\rangle_{W^{l,p}(K)}\right)^q \tag{11}$$

$$= N^{\frac{(l-m)q}{d}}\sum_K |K|^{1+\frac{(l-m)q}{d}} \cdot \left(1 + \alpha^{-1}\langle u\rangle_{W^{l,p}(K)}\right)^q$$

$$= N^{\frac{d+q(l-m)}{d}} \cdot \frac{1}{N}\sum_K \left[|K| \cdot \left(1 + \alpha^{-1}\langle u\rangle_{W^{l,p}(K)}\right)^{\frac{dq}{d+q(l-m)}}\right]^{\frac{d+q(l-m)}{d}}.$$

$$\geq N^{\frac{d+q(l-m)}{d}} \cdot \left[\frac{1}{N}\sum_K |K| \cdot \left(1 + \alpha^{-1}\langle u\rangle_{W^{l,p}(K)}\right)^{\frac{dq}{d+q(l-m)}}\right]^{\frac{d+q(l-m)}{d}} \tag{12}$$

$$= \left[\sum_K |K| \cdot \left(1 + \alpha^{-1}\langle u\rangle_{W^{l,p}(K)}\right)^{\frac{dq}{d+q(l-m)}}\right]^{\frac{d+q(l-m)}{d}}.$$

Since the equality in (11) holds when

$$\frac{1}{d}\operatorname{tr}\left((F'_K)^{-1}(F'_K)^{-T}\right) = \det\left((F'_K)^{-1}(F'_K)^{-T}\right),$$
$$\frac{1}{d}\operatorname{tr}\left((F'_K)^T(F'_K)\right) = \det\left((F'_K)^T(F'_K)\right),$$

from Lemma 1.3, $M$ is of the form
$$M_K \equiv \theta_K I.$$
for some scalar function $\theta = \theta_K$. The equidistribution condition implies that
$$\rho_K |K| = C_1, \quad \forall K \in \mathscr{T}$$
for some constant $C_1$. Noticing that the equality in (12) holds when
$$|K| \cdot (1 + \alpha^{-1} \langle u \rangle_{W^{l,p}(K)})^{\frac{dq}{d+q(l-m)}} = C, \quad \forall K \in \mathscr{T}_h,$$
for some constant $C$, we can get that
$$\rho_K \equiv \left(1 + \alpha^{-1} \langle u \rangle_{W^{l,p}(K)}\right)^{\frac{dq}{d+q(l-m)}}, \quad \forall K \in \mathscr{T},$$
$$M_K \equiv \left(1 + \alpha^{-1} \langle u \rangle_{W^{l,p}(K)}\right)^{\frac{2q}{d+q(l-m)}} I, \quad \forall K \in \mathscr{T}.$$

Then we take the error of approximating $\rho$ into consideration, since it can not be ignored in practice. And we provide the interpolation error bound involving the error of approximating $\rho$.

**Theorem 1** Choose monitor function as Proposition 1. Suppose the error of approximating $\rho$ is
$$\|\rho - \tilde{\rho}\|_{L_{\frac{d}{d+q(l-m)}}(\Omega)} < \epsilon,$$
then the interpolation error of the $M$-uniform mesh $\mathscr{T}$ is
$$|u - \Pi_k u|_{W^{m,q}(\Omega)} \le C\alpha N^{-\frac{(l-m)}{d}} (\sigma^{\frac{d+q(l-m)}{dq}} + \epsilon^{\frac{d+q(l-m)}{dq}}),$$
where $\sigma = \sum_K |K| \rho_K$.

Proof: In this case,
$$\tilde{\rho}_K |K| = C_1, \quad \forall K \in \mathscr{T}.$$
Suppose $\epsilon_K = \|\rho_K - \tilde{\rho}_K\|_{L_{\frac{d}{d+q(l-m)}}(K)}$ ($\forall K \in \mathscr{T}$), then
$$E(\mathscr{T}(M)) = N^{\frac{d+q(l-m)}{d}} \cdot \frac{1}{N} \sum_K [|K| \cdot (\tilde{\rho}_K - \rho_K + \rho_K)]^{\frac{d+q(l-m)}{d}}$$
$$= N^{\frac{d+q(l-m)}{d}} \cdot \frac{1}{N} \sum_K [|K| \cdot (\tilde{\rho}_K - \rho_K) + C_1]^{\frac{d+q(l-m)}{d}}$$
$$\le N^{\frac{d+q(l-m)}{d}} \cdot \frac{1}{N} \sum_K (|K| \cdot \epsilon_K + C_1)^{\frac{d+q(l-m)}{d}}.$$

Because $0 \le m \le l$ and $0 \le q, d$, the exponent $\frac{d+q(l-m)}{d} \ge 1$. Consequently,
$$E(\mathscr{T}(M)) \le N^{\frac{d+q(l-m)}{d}} \cdot \frac{1}{N} \sum_K \left[ C_1^{\frac{d+q(l-m)}{d}} + \frac{d+q(l-m)}{d} (|K| \cdot \epsilon_K + C_1)^{\frac{q(l-m)}{d}} \epsilon_K \right]$$
$$\le N^{\frac{d+q(l-m)}{d}} \cdot \left[ \frac{1}{N} \sum_K C_1 \right]^{\frac{d+q(l-m)}{d}}$$
$$+ N^{\frac{d+q(l-m)}{d}} \cdot \frac{1}{N} \sum_K \frac{d+q(l-m)}{d} (|K| \cdot \epsilon_K + C_1)^{\frac{q(l-m)}{d}} \epsilon_K$$
$$\le N^{\frac{d+q(l-m)}{d}} \cdot \left[ \frac{1}{N} \sum_K C_1 \right]^{\frac{d+q(l-m)}{d}} + N^{\frac{d+q(l-m)}{d}} \cdot \frac{1}{N} \sum_K \left[ (C_2 \epsilon_K)^{\frac{d}{d+q(l-m)}} \right]^{\frac{d+q(l-m)}{d}}$$
$$= N^{\frac{d+q(l-m)}{d}} \cdot \left[ \frac{1}{N} \sum_K C_1 \right]^{\frac{d+q(l-m)}{d}} + N^{\frac{d+q(l-m)}{d}} \cdot \left[ \frac{1}{N} \sum_K (C_2 \epsilon_K)^{\frac{d}{d+q(l-m)}} \right]^{\frac{d+q(l-m)}{d}}$$
$$= \left[ \sum_K |K| \cdot \left(1 + \alpha^{-1} \langle u \rangle_{W^{l,p}(K)}\right)^{\frac{dq}{d+q(l-m)}} \right]^{\frac{d+q(l-m)}{d}} + C_2 \left( \sum_K \epsilon_K^{\frac{d}{d+q(l-m)}} \right)^{\frac{d+q(l-m)}{d}}.$$

Since

$$\left(\sum_K \epsilon_K^{\frac{d}{d+q(l-m)}}\right)^{\frac{d+q(l-m)}{d}} = \left(\sum_K \int_K |\rho_K - \tilde{\rho_K}|^{\frac{d}{d+q(l-m)}} dx\right)^{\frac{d+q(l-m)}{d}}$$

$$= \left(\int_\Omega |\rho - \tilde{\rho}|^{\frac{d}{d+q(l-m)}} dx\right)^{\frac{d+q(l-m)}{d}}$$

$$= ||\rho - \tilde{\rho}||_{L_{\frac{d}{d+q(l-m)}}(\Omega)}$$

$$< \epsilon,$$

we finally have

$$|u - \Pi_k u|_{W^{m,q}(\Omega)} \le C\alpha N^{-\frac{(l-m)}{d}} \left(\sigma^{\frac{d+q(l-m)}{dq}} + \epsilon^{\frac{d+q(l-m)}{dq}}\right).$$

The theorem shows that the error bound increases as $\epsilon$ increases. Besides, there is a trade-off between $\alpha$ and $\sigma$. To examine how to choose $\alpha$, we then provide some lemmas.

**Lemma 2.1** For any real number $t > 0$

$$|a + b|^t \le c_t(|a|^t + |b|^t), \quad \forall a, b \in \mathbb{R},$$

$$|a|^t - c_t|b|^t \le c_t|a - b|^t, \quad \forall a, b \in \mathbb{R},$$

where

$$c_t = 2^{\max\{t-1, 0\}}.$$

Moreover, if $t \in (0, 1)$, then it holds that

$$\left||a|^t - |b|^t\right| \le |a - b|^t, \quad \forall a, b \in \mathbb{R}.$$

**Lemma 2.2** Suppose that the assumptions in Lemma 1.1 and Theorem 1 hold,

$$\left[\sum_K |K| \cdot \langle u \rangle_{W^{1,p}(K)}^{\frac{dq}{d+q(l-m)}}\right]^{\frac{d+q(l-m)}{d}} \to \left[\int_\Omega \|D^l u\|_{l^p}^{\frac{dq}{d+q(l-m)}} d\mathbf{x}\right]^{\frac{d+q(l-m)}{d}} \tag{13}$$

as long as

$$h = \max_K h_K \to 0 \quad \text{as} \quad N \to \infty,$$

where $h_K$ is the diameter of $K$.

**Lemma 2.3 (Poincaré's inequality for a convex domain; $u_\Omega$)** Let $\Omega \subset \mathbb{R}^d$ be a bounded convex domain with diameter $h_\Omega$. Then, for any $1 \le q \le p < \infty$,

$$\|u - u_\Omega\|_{L^q(\Omega)} \le c_{p,q} h_\Omega |\Omega|^{\frac{1}{q} - \frac{1}{p}} \|\nabla u\|_{L^p(\Omega)}, \quad \forall u \in W^{1,p}(\Omega),$$

where $c_{p,q}$ is a constant depending only on $p$ and $q$ which has values or bounds as follows:

(2) If $1 < q \le p < \infty$,

$$c_{p,q} \le q^{\frac{1}{q}} \left(\frac{p}{p-1}\right)^{\frac{1}{p} - \frac{1}{q}} 2^{\frac{p-1}{p}}.$$

(2) If $q = 1 < p < \infty$,

$$\left(\int_0^1 (x(1-x))^{\frac{p}{p-1}} dx\right)^{\frac{p-1}{p}} \le c_{p,1} \le 2 \left(\int_0^1 (x(1-x))^{\frac{p}{p-1}} dx\right)^{\frac{p-1}{p}}.$$

(3) If $p = q = 1$, $c_{1,1} = \frac{1}{2}$.

(4) If $p = q = 2$, $c_{2,2} = \frac{1}{\pi}$.

**Lemma 2.4** For any real numbers $\gamma \in (0, 1]$ and $p \in [1, \infty)$ and any mesh $\mathscr{T}$ for $\Omega$,

$$\|\nu\|_{L^{jp}(\Omega)}^{\gamma p} \leq \sum_K |K|^{1-\gamma} \|\nu\|_{L^{jp}(K)}^{\gamma p}$$

$$\leq c_{\gamma p}^2 \|\nu\|_{L^{jp}(\Omega)}^{\gamma p} + c_{\gamma p}(1 + c_{\gamma p}) c_{p,p}^{\gamma p} h^{\gamma p} |\Omega|^{1-\gamma} \|\nabla \nu\|_{L^p(\Omega)}^{\gamma p}$$

for any $\nu \in W^{1,p}(\Omega)$, where $h = \max_{K \in \mathscr{T}} h_K$ and the constants $c_{\gamma p}$ and $c_{p,p}$ are defined in Lemma 2.1 and Lemma 2.3, respectively.

**Lemma 2.5** For any real numbers $\gamma \in (0, 1]$ and $p \in [1, \infty)$ and any mesh $\mathscr{T}$ for $\Omega$,

$$\|v\|_{L^{\gamma p}(\Omega)}^{\gamma p} \leq \sum_K |K|^{1-\gamma} \|\nu\|_{L^p(K)}^{\gamma p} \leq |\Omega|^{1-\gamma} \|\nu\|_{L^p(\Omega)}^{\gamma p}, \quad \forall \nu \in L^p(\Omega).$$

**Lemma 2.6** For an $M$-uniform mesh $\mathscr{T}$,

$$h_K \leq \hat{h} \left(\frac{\sigma}{N}\right)^{\frac{1}{d}} \frac{1}{\sqrt{\lambda_{min}}}, \quad \forall K \in \mathscr{T},$$

where $\hat{h} = 2\sqrt{\frac{d}{d+1}} \left(\frac{d!}{\sqrt{d+1}}\right)^{\frac{1}{d}}$ and $\lambda_{min}$ is the minimum eigenvalue of $M_K$.

Two different choices of $\alpha$ are examined. The first choice is taking $\alpha \to 0$, which means that $M$ is dominated by $\langle u \rangle_{W^{l,p}(K)}$. For the second choice, $\alpha$ is taken such that $\sigma$ is bounded by a constant while keeping $M$ invariant under a scaling transformation of $u$.

**Theorem 2** Suppose that the assumptions in Lemma 1.1 hold and that $q \leq p$. Suppose also that the mesh density function and the monitor function are chosen as in Proposition 1.
(a) Choose $\alpha$ such that $\alpha \to 0$, if there exists a constant $\beta$ such that

$$\|D^l u(\mathbf{x})\|_{l^p} \geq \beta > 0, \quad \text{a.e. in } \Omega$$

holds to ensure that $M$ is positive definite, then for $u \in W^{l,p}(\mathbf{\Omega})$

$$|u - \Pi_k u|_{W^{m,q}(\Omega)} \leq C N^{-\frac{(l-m)}{d}} \left(\sum_K |K| \langle u \rangle_{W^{l,p}(K)}^{\frac{dq}{d+q(l-m)}}\right)^{\frac{d+q(l-m)}{dq}},$$

with

$$\lim_{N \to \infty} \left(\sum_K |K| \langle u \rangle_{W^{l,p}(K)}^{\frac{dq}{d+q(l-m)}}\right)^{\frac{d+q(l-m)}{dq}} \leq C|u|_{W^{l,\frac{dq}{d+q(l-m)}}(\Omega)}.$$

If further $u \in W^{l+1,p}(\mathbf{\Omega})$, then

$$|u - \Pi_k u|_{W^{m,q}(\mathbf{\Omega})}$$

$$\leq C N^{-\frac{(l-m)}{d}} \left(|u|_{W^{l,\frac{dq}{d+q(l-m)}}(\mathbf{\Omega})} + N^{-\frac{1}{d}} |u|_{W^{l,p}(\mathbf{\Omega})}^{\frac{q}{d+q(l-m)}} |u|_{W^{l+1,p}(\mathbf{\Omega})}\right).$$

(b) Choose

$$\alpha \equiv \left[\frac{1}{|\Omega|} \sum_K |K| \langle u \rangle_{W^{l,p}(K)}^{\frac{dq}{d+q(l-m)}}\right]^{\frac{d+q(l-m)}{dq}},$$

then for $u \in W^{l,p}(\Omega)$,

$$|u - \Pi_k u|_{W^{m,q}(\Omega)} \leq C N^{-\frac{(l-m)}{d}} \left(\sum_K |K| \langle u \rangle_{W^{l,p}(K)}^{\frac{dq}{d+q(l-m)}}\right)^{\frac{d+q(l-m)}{dq}} (1 + \epsilon^{\frac{d+q(l-m)}{dq}}),$$

with

$$\lim_{N \to \infty} \left(\sum_K |K| \langle u \rangle_{W^{l,p}(K)}^{\frac{dq}{d+q(l-m)}}\right)^{\frac{d+q(l-m)}{dq}} \leq C|u|_{W^{l,\frac{dq}{d+q(l-m)}}(\Omega)}$$

holding. If further $u \in W^{l+1,p}(\mathbf{\Omega})$, then

$$|u - \Pi_k u|_{W^{m,q}(\Omega)} \le CN^{-\frac{(l-m)}{d}} \left( |u|_{W^{l,\frac{dq}{d+q(l-m)}}(\Omega)} + N^{-\frac{1}{d}} |u|_{W^{l+1,p}(\Omega)} \right) (1 + \epsilon^{\frac{d+q(l-m)}{dq}}).$$

Proof: (a) Since the equidistribution and alignment conditions are invariant under a scaling transformation of $M_K$, let

$$M_K = \alpha^{\frac{2q}{d+q(l-m)}} M_K^{ixo} = \left( \alpha + \langle u \rangle_{W^{l,p}(K)} \right)^{\frac{2q}{d+q(l-m)}} I, \quad \forall K \in \mathscr{T},$$

so the corresponding mesh density function is

$$\rho_K = \alpha^{\frac{dq}{d+q(l-m)}} \rho_K = \left( \alpha + \langle u \rangle_{W^{l,p}(K)} \right)^{\frac{dq}{d+q(l-m)}}, \quad \forall K \in \mathscr{T}.$$

As $\alpha^{\frac{dq}{d+q(l-m)}} \to 0$,

$$||\rho - \tilde{\rho}||_{L_{\frac{d}{d+q(l-m)}}(\Omega)} \to 0,$$

$$\rho_K \to \langle u \rangle_{W^{l,p}(K)}^{\frac{dq}{d+q(l-m)}}.$$

By the same argument as Proposition 1, we get

$$|u - \Pi_k u|_{W^{m,q}(\mathbf{\Omega})} \le CN^{-\frac{(l-m)}{d}} \left( \sum_K |K| \langle u \rangle_{W^{l,p}(\boldsymbol{K})}^{\frac{dq}{d+q(l-m)}} \right)^{\frac{d+q(l-m)}{dq}}. \tag{14}$$

From Lemma 2.5 and Lemma 2.6,

$$h_K \le CN^{-\frac{1}{d}} \lambda_{min}(M_K)^{-\frac{1}{2}} \sigma^{\frac{1}{d}}$$

$$\le CN^{-\frac{1}{d}} \beta^{-\frac{1}{2}} \left( \sum_K |K| \langle u \rangle_{W^{l,p}(K)}^{\frac{dq}{d+q(l-m)}} \right)^{\frac{1}{d}}$$

$$\le CN^{-\frac{1}{d}} \boldsymbol{\beta}^{-\frac{1}{2}} |\boldsymbol{u}|_{W^{l,p}(\mathbf{\Omega})}^{\frac{q}{d+q(l-m)}}, \tag{15}$$

which means that this $M$-uniform mesh satisfies the condition of Lemma 2.2. And from Eq(13), it is obvious that

$$\lim_{N \to \infty} \left( \sum_K |K| \langle u \rangle_{W^{l,p}(K)}^{\frac{dq}{d+q(l-m)}} \right)^{\frac{d+q(l-m)}{dq}} \le C|u|_{W^{l,\frac{dq}{d+q(l-m)}}(\Omega)}.$$

For $u \in W^{l+1,p}(\mathbf{\Omega})$, this limit can be refined from Lemma 2.4 and Eq(15) to

$$\left( \sum_K |K| \langle u \rangle_{W^{l,p}(K)}^{\frac{dq}{d+q(l-m)}} \right)^{\frac{d+q(l-m)}{dq}}$$

$$= \left[ \sum_K |K|^{1-\frac{dq}{p(d+g(l-m))}} |u|_{W^{l,p}(K)}^{\frac{dq}{d+q(l-m)}} \right]^{\frac{d+q(l-m)}{dq}}$$

$$\le C \left[ |u|^{\frac{dq}{d+q(l-m)}} + h^{\frac{dq}{d+q(l-m)}} |u|_{W^{l,1,p'}(\Omega)}^{\frac{d+q(l-m)}{dq}} \right]^{\frac{d+q(l-m)}{dq}}$$

$$\le C \left[ |u|_{W^{l,\frac{dq}{d+q(l-m)}}(\Omega)} + N^{-\frac{1}{d}} |u|_{W^{l,p'}(\Omega)}^{\frac{q}{d+q(l-m)}} |u|_{W^{l+1,p}(\Omega)}^{\frac{q}{d+q(l-m)}} \right].$$

Inserting this into Eq(14) gives

$$|u - \Pi_k u|_{W^{m,q}(\boldsymbol{\Omega})}$$

$$\leq C N^{-\frac{(l-m)}{d}} \left( |u|_{W^{l, \frac{dq}{d+q(l-m)}}(\boldsymbol{\Omega})} + N^{-\frac{1}{d}} |u|_{W^{l,p}(\boldsymbol{\Omega})}^{\frac{q}{d+q(l-m)}} |u|_{W^{l+1,p}(\boldsymbol{\Omega})} \right).$$

(b) $\sigma \leq C$ for some constant:

$$\sigma = \sum_K |K| \left[ 1 + \alpha^{-1} \langle u \rangle_{W^{l,p}(K)} \right]^{\frac{dq}{d+q(l-m)}}$$

$$\leq c_{\frac{dq}{d+q(l-m)}} \sum_K |K| \left[ 1 + \alpha^{-\frac{dq}{d+q(l-m)}} \langle u \rangle_{W^{l,p}(K)}^{\frac{dq}{d+q(l-m)}} \right]$$

$$= c_{\frac{dq}{d+q(l-m)}} \left[ |\Omega| + \alpha^{-\frac{dq}{d+q(l-m)}} \sum_K |K| \langle u \rangle_{W^{l,p}(K)}^{\frac{dq}{d+q(l-m)}} \right]$$

$$= 2|\Omega| c_{\frac{dq}{d+q(l-m)}},$$

where constant $c_{\frac{dq}{d+q(l-m)}}$ is defined as Lemma 2.1. From this and Theorem 1, we get

$$|u - \Pi_k u|_{W^{m,q}(\Omega)} \leq C N^{-\frac{(l-m)}{d}} \left[ \sum_K |K| \langle u \rangle_{W^{l,p}(K)}^{\frac{dq}{d+q(l-m)}} \right]^{\frac{d+q(l-m)}{dq}} (1 + \epsilon^{\frac{d+q(l-m)}{dq}}).$$

From the definition of $\alpha$,

$$|\Omega|(\alpha)^{\frac{dq}{d+q(l-m)}} = \sum_K |K| \langle u \rangle_{W^{l,p}(K)}^{\frac{dq}{d+q(l-m)}} = \sum_K |K|^{1 - \frac{dq}{p(d+q(l-m))}} |u|_{W^{l,p}(K)}^{\frac{dq}{d+q(l-m)}}.$$

Lemma 2.4 and the above equation give

$$|u|_{W^{l, \frac{dq}{d+q(l-m)}}(\Omega)}^{\frac{dq}{d+q(l-m)}} \leq |\Omega|(\alpha)^{\frac{dq}{d+q(l-m)}}$$

$$\leq C \left( |u|_{W^{l, \frac{dq}{d+q(l-m)}}(\Omega)}^{\frac{dq}{d+q(l-m)}} + N^{-\frac{q}{d+q(l-m)}} |u|_{W^{l+1,p}(\Omega)}^{\frac{dq}{d+q(l-m)}} \right).$$

Combining these with Theorem 1, we obtain

$$|u - \Pi_k u|_{W^{m,q}(\Omega)} \leq C N^{-\frac{(l-m)}{d}} \left( |u|_{W^{l, \frac{dq}{d+q(l-m)}}(\Omega)} + N^{-\frac{1}{d}} |u|_{W^{l+1,p}(\Omega)} \right) (1 + \epsilon^{\frac{d+q(l-m)}{dq}}).$$

From the theorem, we can infer that the error bound decreases with the increase of $N$, and increases with the increase with the increase of $\epsilon$ in case (b). Intuitively speaking, because the term that contains $\epsilon$ in Theorem 1's error bound is $C\alpha N^{-\frac{l-m}{dq}} \epsilon^{\frac{d+q(l-m)}{dq}}$, as $\alpha \to 0$, $\epsilon$ does not influence the error bound in case (a). Consequently, taking $\alpha \to 0$ is more suitable when the error of approximating $\rho$ is non-negligible.

## G  MONITOR FUNCTION

### G.1  FORM

We decide the index $l, m, p, q$ following the assumptions in Proposition 1: the interpolated function $u \in W^{l,p}(K)$ and error norm $e \in W^{m,q}(K)$. Noticing that we treat the discrete input $u$ as a piecewise constant function and we take MSE as the error norm, we choose $l = 1, m = 0, p = q = 2$, so the monitor function for 2-D settings is

$$M = (1 + \|\nabla u\|_{l_2}/\alpha)^1 I.$$

As for $\alpha$, we choose

$$\alpha = \frac{1}{100|\Omega|} \sum_K |K| \langle u \rangle_{W^{1,2}(K)}$$

to let $\alpha$ be as close to case (a) as possible and make $M$ still invariant under a scaling transformation of $u$. The constant 100 is chosen with parameter tuning as shown in Appendix I.

## G.2 Calculation

The form of the monitor function needs to take the derivative of the discrete $u$. For rectangular original mesh, we take the one-order finite difference. For the triangular mesh, we first extrapolate $u$ to get a continuous function and then get the derivative with automatic gradient computation in Pytorch (Paszke et al., 2017). Adopting the way used in GEN (Alet et al., 2019), we extrapolate $u = (u_l)$ as

$$u(y) = \sum_l r(y, x_l) u_l^T \tag{16}$$

over the domain $\Omega$. $r(x) = softmax(-C\|x - y\|)$ is a soft nearest neighbor representation function, where $C$ is a hyper-parameter. In practice, $C$ is chosen as $\lfloor \sqrt{N} \rfloor$.

# H Model Architecture

## H.1 DMM

Encoder 1 is used to encode state $u$. For the Burgers' equation, we take a 2-D CNN for the regular rectangular mesh:

- 1 input channel, 8 output channels, 5×5 kernel,
- 8 input channel, 16 output channels, 5×5 kernel,
- 16 input channel, 8 output channels, 5×5 kernel,
- 8 input channel, 5 output channels, 5×5 kernel,

followed by a flatten operation and two linear layers with latent dimensions 1024 and 512. Residual connections are used between 2-D convolution layers. And the nonlinear activation function is the tangent function.

For flow around a cylinder, the mesh is triangular, so we take the model architecture as follows:

- embedding MLP: a linear layer from dimension 3 to latent dimension 4 → a 1-D batch norm layer → tangent activation function → a linear layer from latent dimension 4 to latent dimension 4 → a 1-D batch norm layer,
- three message-passing layers with 4 latent features and tangent activation function,
- decoding MLP: linear layers from dimension 4 to 128 to 1.

Encoder 2 is used to encode coordinates $(t, x)$ and taken as MLP with tangent activation function. For the Burgers' equation, the layers' numbers of latent nodes are 32 and 512, while in the second experiment are 16 and 512.

The decoder is the same for two experiments. We take two linear layers with tangent activation function from latent dimension 1024 to 512 and then to 1.

## H.2 MM-PDE

MM-PDE's model architecture is the same for the two settings. $\mathscr{G}_1$ and $\mathscr{G}_2$ are the same referring to MP-PDE (Brandstetter et al., 2022). There is an MLP encoder, 6 message-passing layers, and an MLP decoder.

The encoder gives every node $i$ a new node embedding, which is computed as

$$n_i^0 = Encoder([u_i, x_i, t]),$$

where $u_i$ is the solution's value at node $i$ and $x_i$ is the space coordinate. Then the embedding is inputted into message-passing layers. Every layer has 128 hidden features, and includes a 2-layer edge update MLP $EU$ and a 2-layer node update MLP $NU$. For the $m$-th step of learned message passing, the computation process of the edge update network from node $j$ to node $i$ is

$$m_{ij}^m = EU\left(n_i^m, \mathbf{n}_j^m, u_i - u_j, x_i - x_j\right).$$

And the computation process of the node update network on node $i$ is

$$n_i^{m+1} = NU \left( n_i^m, \sum_{j \in \mathcal{N}(i)} m_{ij}^m \right),$$

where $\mathcal{N}_i$ is the neighbors of node $i$. Here we pick the nearest $K$ nodes of the node $i$ as its neighbors $\mathcal{N}_i$, and $K$ is set to be 35. At last, a shallow 1D convolutional network is used, which shares weights across spatial locations.

In the framework $I$, $Itp_1$ and $Itp_2$ have the same architecture: two linear layers with latent dimensions 128 and 64 respectively. The input of them is the concatenation of its position and its nearest $K$ neighbors' positions, where $K = 30$. Then its output is $K$ weights $w_1, \cdots, w_K$, and the final interpolation result is the weighted sum of neighbors' values. As for the residual cut network, it is a 4-layer MLP with latent dimensions from 2048 to 512 to 2048 then to the original dimension.

# I  ADDITIONAL EXPERIMENT RESULTS

## I.1  HYPERPARAMETER SEARCH

We conduct experiments for selecting optimal key hyperparameters. The results are presented in Table 12, from which we choose $\beta = 1000, K_{itp} = 30, K_{mp} = 35$.

Table 12: Hyperparameter search for the Burgers' equation. $\beta$ is the weight of $l_bound$ in neural moving mesh's physics loss. $K_{itp}$ and $K_{mp}$ means the number of neighbors for interpolation and message passing used in MM-PDE respectively.

| HYPERPARAMETER | ERROR |
|---|---|
| $\beta = 1000, K_{itp} = 30, K_{mp} = 35$ | **6.63e-07** |
| $\beta = 100, K_{itp} = 30, K_{mp} = 35$ | 1.02e-06 |
| $\beta = 10000, K_{itp} = 30, K_{mp} = 35$ | 1.01e-06 |
| $\beta = 1000, K_{itp} = 35, K_{mp} = 35$ | 1.23e-06 |
| $\beta = 1000, K_{itp} = 25, K_{mp} = 35$ | 6.67e-07 |
| $\beta = 1000, K_{itp} = 30, K_{mp} = 35$ | 2.51e-05 |
| $\beta = 1000, K_{itp} = 30, K_{mp} = 35$ | 9.41e-07 |

## I.2  CHOOSE $\alpha$

As explained before, the form of $\alpha$ is

$$\alpha = \frac{1}{C|\Omega|} \sum_K |K| \langle u \rangle_{W^{1,2}(K)} .$$

To decide the value of hyperparameter $C$, we try $C = 10, 100, 1000$ respectively. From Table 13, we finally choose $C = 100$ whose error is lowest.

Table 13: Choose $C$ in the monitor function.

| $C$ | ERROR |
|---|---|
| 10 | 7.14e-07 |
| 100 | **6.63e-07** |
| 1000 | 1.02e-06 |

