# OpenReview forum: "Better Neural PDE Solvers Through Data-Free Mesh Movers"
_ICLR.cc/2024/Conference — ICLR 2024 poster_

### Official Review · Reviewer_Uu92 · 2023-10-28

**Soundness:** 2 fair
**Presentation:** 2 fair
**Contribution:** 3 good
**Rating:** 6
**Confidence:** 4

**Summary:**

The authors proposed a mesh mover based PDE solver in this paper. The solver consists of two main components: a mesh mover and a neural PDE solver. The mesh mover is trained unsupervised based on a physical loss motivated by optimal-transport based Monge-Amp`ere mesh movement method. The GNN neural PDE solver is trained supervised given solutions from numerical simulations. The experiments show that the proposed method can outperform existing NN solvers on 2D Burger's equation and flow past cylinder cases.

**Strengths:**

- The idea to design a neural PDE solver with mesh movement method is novel.
- The Monge-Amp`ere equation motivated physics loss for mesh mover training is intuitive.
- The use of BFGS for memory intensive training and the sampling method for Monge-Ampere equation are sound.
- The background part is informative and help understand the proposed method.

**Weaknesses:**

- One essential challenge for r-adaptive (i.e., mesh movement) based mesh adaptive methods is mesh tangling. The mesh tangling happens when the mesh have elements with negative Jacobian-determinant, or in other words, the edges of mesh nodes come across each other after being moved. Although the optimal-transport based Monge-Amp`ere mesh movement method is designed to alleviate the mesh tangling issue, it is hard to guarantee tangling-free especially for a learned model, which only has a soft physical loss as the constraint. It is unclear how the proposed DMM perform regarding mesh tangling as there are no related evaluations in the paper.
- The Monge-Amp`ere based physical loss is the key contribution of this paper, however it is not clear how each part of the loss i.e. $loss_{equation}$, $loss_{bound}$ and $loss_{convex}$ contribute to the model performance.
- The experiments are limited to one fixed mesh resolution (48 x 48 or 2521 triangular lattices) and there are no evaluations on generalization of the proposed model, i.e., trained on a (small) resolution and test on other (potentially larger) resolution. Therefore, it is not clear how the proposed method perform given unseen data / boundary conditions/ physical parameters in equations.
- The presentation and text of the paper should be improved.

**Questions:**

- The mesh tangling is a key challenge for mesh movement methods. It is hard to evaluate the performance of the proposed mesh mover without the evaluations on this potential issue. I would suggest that adding a solid discussion for mesh tangling will improve the soundness of this paper.
- Regarding to the physical loss, the $loss_{equation}$ part is designed based on the difference between the current mesh and the uniform mesh. In a time-dependent problem, does it indicate the mesh mover always move the mesh nodes starting from the very initial uniform mesh without using the moved mesh in the last timestep? Does this strategy perform better comparing to reuse the previous moved mesh as the starting mesh?
- An ablation study for contributions of each components in the physical loss ($loss_{equation}$, $loss_{bound}$ and $loss_{convex}$) will help evaluate its effectiveness.
- Mesh movement methods may struggle with cases where the monitor functions give large values near boundaries, e.g. the peak values of the initial conditions of the burger's equation are located close to simulation domain boundary. It would be more convincing if there are such test cases for evaluating the proposed model.
- It is mentioned in the paper that the mesh for Burger's is 48x48, however it is only 20x20 shown in Figure 2. Are there any missing results for the 48 x 48 resolution?

Minors:
- The DMM model architecture is not shown in Figure 2(a)

---

> ### Author Response · Authors · 2023-11-20
> **Response to Reviewer Uu92**
>
> We sincerely appreciate your mention of the strengths of our paper and your pieces of advice which help us enhance the manuscript.
>
> > **Suggestion about the mesh tangling.**
>
> **Answer**: Thanks for your suggestion.
>
> - Firstly, as we use the output of DMM to enhance neural solvers' message passing rather than use it as generated mesh for traditional PDE solvers, mesh tangling does not pose a problem in our context.
> - Secondly, to address your concern, we also made an additional discussion and exploration on it in Section 6.4 and Appendix C.2.  Constraining the Jacobian determinant of $f$ to be non-negative is equivalent to requiring $\Phi$ in our paper to be convex, which is already constrained by $l_{convex}$. The trend of $l_{convex}$ during training is illustrated in Figure 3 in our paper, showing that it can reach a very low level (nearly zero) early in the training process. Additionally, we find that the generated grid points of test data have $l_{convex}=0$. These demonstrate that this constraint is well met.
>
> > **Question about roles of three losses within the physics loss.**
>
> **Answer**: Thanks for your question. We conduct new experiments to explore how the three losses impact the performance of DMM in Section 6.4 and Appendix C.3. We reduce the weights of the three losses within the physics loss by a factor of ten for training respectively. Results show that when the weight of $l{equation}$ is decreased, the std and range increase the most, suggesting that the MA equation is not adequately satisfied. The result does not deviate significantly from the original when the weight of $l_{convex}$ is decreased, as $l_{convex}$ can already reach a low level as stated in Answer to Re 1. Figure 4 in the paper shows the plotted meshes of decreased $l_{bound}$, implying that boundary conditions are not as well satisfied as before. All the results above conform to intuition.
>
>
>
> > **Question about resolution transformation.**
>
> **Answer**: Thanks for your question. One of DMM's advantages is its ability to generate meshes of varying resolutions during inference.
>
> - We have added a new experiment in Section 6.4 and Appendix C.1. The result shows that DMM can handle data of resolution different from the training data's resolution.
> - Please note that since the selected encoder 1 cannot handle varying resolutions, we need to interpolate the state $u$ prior to inputting states of different resolutions. We assess bicubic interpolation and the softmax interpolation as in Eq 16, with training data at $48\times48$ and 2521 resolutions.
> - The meshes from the pre-trained DMM are illustrated in Figure 2 in the paper, and associated quantitative results are in Table 7 and 8 in the paper. Results show that the data adheres well to the MA equation regardless of resolution.
>
>
>
> > **Question about how the proposed method performs given unseen data/boundary conditions/physical parameters in equations.**
>
> **Answer**: Thanks for your question. Firstly, our method can generalize to unseen data in a considerable range due to the generalization ability of the neural networks and the physics-informed data-free DMM training. However, we don't think that our current model can generalize to different boundary conditions and physical parameters in equations since we have not taken it into the input of the model.
>
>
>
> > **Suggestion of improving presentation and text.**
>
> **Answer:** Thanks for your suggestion. We have modified our paper according to the reviewers' suggestions and revised the entire paper.
>
>
>
> >  **Question about moving the mesh nodes using the moved mesh in the last timestep.**
>
> **Answer**: Thanks for your question. The reason why we can not learn the meshes based on meshes in the last timestep is that the meshs in the last timestep are not known explicitly at the DMM training period, which means that we can not deduce the physics loss in this situation.
>
>
>
> > **Question about cases where the monitor functions give large values near boundaries.**
>
> **Answer**: Thanks for your question. A new visualization of these cases is provided in Appendix D. We choose some states where the maximum value of the monitor function is relatively close to the boundary to generate meshes. Additionally, we have generated new states with maximum values much closer to boundaries and obtained the new mesh. We can see from Figure 7 that DMM can also handle such situations.
>
>
> > **Question about why to only show 20x20 meshes for Burgers' equation.**
>
> **Answer**: Thanks for your question.  We have added 48x48 meshes to Appendix D.
> The reason why we show 20x20 meshes previously is that 20x20 meshes are clearer and can help reduce the length of the main text.
>
>
> > **Suggestion about typos of Figure 2(a).**
>
> **Answer**: Thanks for your suggestion. We have fixed it and revised the manuscript to make the paper more clear.

---

> ### Author Response · Authors · 2023-11-23
> **We have made efforts to address your concerns. Are there any other suggestions?**
>
> Thank you once again for your valuable insights and suggestions.
>
> We have meticulously addressed your concerns. Please let us know if you have any further suggestions.

---

### Official Review · Reviewer_W4Uo · 2023-10-30

**Soundness:** 3 good
**Presentation:** 2 fair
**Contribution:** 3 good
**Rating:** 6
**Confidence:** 3

**Summary:**

The authors propose a mesh moving method to improve dynamics learning of graph neural networks. Theoretical understanding of the neural mesh adapter is provided, and empirical results are shown on two problems: the Burgers' equation and flow past cylinder, both in 2D.

**Strengths:**

A detailed explanation was provided for the framework, with plenty of background information. The method applies well in practice, and the appendix provides enough technical detail on results in the main paper.

**Weaknesses:**

1. Runtimes should be mentioned for the proposed model, especially when comparing with equivalent methods such as GNN, CNN, FNO and LAMP. How much overhead does DMM require, and what number of extra trainable parameters are we talking about in practice?
2. When training the DMM separately, how well is the physics loss being minimized? It is important to know to what extent the equations need to be satisfied until we see an improvement in MM-PDE.
3. A sentence of explanation should be added when training the DMM, where we use a combination of both Adam and BFGS, but only finetune the last layer. Did this show the best performance? Why did Adam alone not suffice. When the high memory consumption is mentioned, what numbers is required by the two example problems shown in this paper?
4. From a practitioner's perspective, it is unclear why the metric for cell volume matters for the examples, aside from showing that the DMM loss was minimized. Most importantly, we never perform any testing on the adapted/moved mesh, hence its structure can be arbitrary as long as it is conducive to the training of the GNN. It has been shown that uniform meshes as a representation perform worse, but is it the case that reducing the cell volume std and range will also monotonically decrease the test loss of the model?
5. For the non-uniform grid example in flow past cylinder, CNN and FNO were not considered. Would the non-uniform equivalent of FNO, the Geometry-Aware FNO GeoFNO, be applicable as a baseline? If so, comparing with non-uniform state-of-the-art methods would strengthen the paper as well.
6. More details should be added on the experimental results, for example, for the flow past cylinder, is the MSE computed on just the velocity, or also pressures? The values reported in Table 2 seem relatively high then, perhaps visualization of the rollout results could also be added to the appendix.

**Questions:**

1. The implications of the ablation study are that having a trainable DMM performs better, but having it trainable also means now it is finetuning on the MSE loss from the solution data, hence the DMM equations may no longer hold. Did you quantify the std and range of cell volumes after this mm+end2end training as well?
2. How are the DMM loss derivatives computed, using finite differences, or autodifferentiability?
3. Is there any limitation in terms of extending the framework to 3D problems? Computational complexity?

---

> ### Author Response · Authors · 2023-11-20
> **Response to Reviewer W4Uo [Part 1]**
>
> Thanks for your valuable suggestions to help us improve the manuscript.
>
> > **Question about the runtimes of MM-PDE and other baselines.**
>
> **Answer**: Thanks for your suggestions. We have added the inference time of MM-PDE to Table 1 in Section 6.1 and list it below.  We can see that MM-PDE is comparable to GNN. But the time is slower CNN and FNO since the base model architecture GNN is slower.
>
> | Model   | MM-PDE (MP-PDE) | MP-PDE | bigger MP-PDE | MM-PDE (MGN) | MGN    | CNN    | FNO    | LAMP   |
> | ------- | --------------- | ------ | ------------- | ------------ | ------ | ------ | ------ | ------ |
> | Time(s) | 0.5192          | 0.3078 | 0.3298        | 0.1187       | 0.0400 | 0.0027 | 0.0159 | 1.4598 |
>
>
>
> > **Question about the overhead DMM requires.**
>
> **Answer**: Thank you for your questions, we have revised Section 6.1 and 6.2 to add the information to the paper for clarity.
>
> - As for the overhead DMM requires, the DMM for the Burgers' equation has a parameter count of 1222738, with a training time of 7.52 hours and memory usage of 15766MB. The DMM for the flow around a cylinder has 2089938 parameters, with a training time of 22.91 hours and memory usage of 73088MB. And the inference of the DMM is 0.001s for the Burgers' equation and 0.088s for the flow around a cylinder, while several tens of seconds are needed for the conventional PDE solver on the Burgers' equation's mesh adaptation. [1] This means that once DMM is trained, the use of it is quite fast and the time exceeds traditional solvers a lot.
>
> [1] Song W, Zhang M, Wallwork J G, et al. M2N: mesh movement networks for PDE solvers[J]. Advances in Neural Information Processing Systems, 2022, 35: 7199-7210.
>
>
>
> > **Question about the physics loss.**
>
> **Answer**: Thank you for your question.
>
> - We have revised Section 6.4 and provided the physics loss of DMM at the beginning and end of the Burgers' equation and flow around a cylinder both below. It is evident that there is a substantial decrease in this loss throughout the training process, indicating that the DMM has been optimized effectively. The result is also listed below.
> - The reason we did not previously provide this loss is that we perceived it as not being sufficiently intuitive. Therefore, we proposed the std and range as more intuitive indicators before.
>
> | Loss           | Init    | Adam    | BFGS    |
> | -------------- | ------- | ------- | ------- |
> | $l$            | 28.248  | 0.045   | 0.026   |
> | $l_{equation}$ | 28.209  | 0.044   | 0.023   |
> | $l_{equation}$ | 3.94e-4 | 1.07e-6 | 2.80e-6 |
> | $l_{convex}$   | 2.47e-7 | 0.0     | 0.0     |
>
> | Loss           | Init    | Adam    | BFGS    |
> | -------------- | ------- | ------- | ------- |
> | $l$            | 0.881   | 0.101   | 0.095   |
> | $l_{equation}$ | 0.547   | 0.095   | 0.093   |
> | $l_{bound}$    | 3.34e-4 | 5.99e-6 | 1.54e-6 |
> | $l_{convex}$   | 0.0     | 0.0     | 0.0     |
>
> > **Question about the optimization of DMM.**
>
> **Answer**: Thanks for your question.
>
> - Our choice to combine the Adam and BFGS optimizers follows many works on Physics-Informed Neural Networks (PINNs) that use several rounds of the LBFGS optimizer at the end of the optimization process [1, 2, 3]. However, in practice, we find that the LBFGS optimizer is quite slow and not very accurate, failing to further reduce the loss. Consequently, we employ the more precise BFGS optimizer.
> - To address your concern, we also have conduced an additional ablation study to show whether the BFGS is effective. The result is also listed in Table 2 in Section 6.4 and the table above. We can conclude that BFGS indeed results in a further reduction of the DMM loss.
> - However, the BFGS optimizer does require more memory, so we only optimize the parameters of the final layer, and we try to increase the width of this layer as much as possible. The current memory usage has already been mentioned earlier: the Burgers' equation requires 15766MB, and the flow around a cylinder requires 73088MB. A single GPU card would not be sufficient to support our training if we optimized the parameters of the last two layers. However, it can be anticipated that optimizing more parameters could potentially improve the optimization results.
>
> [1] Lou Q, Meng X, Karniadakis G E. Physics-informed neural networks for solving forward and inverse flow problems via the Boltzmann-BGK formulation[J]. Journal of Computational Physics, 2021, 447: 110676.
>
> [2] PATRONI R. Towards adaptive PINNs for PDEs: a numerical exploration[J]. 2022.
>
> [3] Pratama D A, Abo-Alsabeh R R, Bakar M A, et al. Solving partial differential equations with hybridized physic-informed neural network and optimization approach: Incorporating genetic algorithms and L-BFGS for improved accuracy[J]. Alexandria Engineering Journal, 2023, 77: 205-226.

---

> > ### Author Response · Authors · 2023-11-20
> > **Response to Reviewer W4Uo [Part 2]**
> >
> > > **Question about the relationship between the cell volume std and range and MM-PDE.**
> >
> > **Answer**: Thanks for your question.
> >
> > - Firstly, the optimal moving mesh is the one satisfying the equidistribution principle under optimal monitor function *M*, which means mesh movement uniformizes cells’ volumes under *M*. Thus we calculate the standard deviation (std) and range of all cells’ volumes as the evaluation metric of DMM.
> >
> > - Secondly, to further show how the std and range of meshes influence the MM-PDE, we conduct a new ablation study in section 6.4. Since the std and range are hard to directly control, we let the weighted sum of identity map $f_I$ and the DMM's coordination transformation map $\tilde{f}$ replace $\tilde{f}$, i.e. $\alpha\tilde{f} + (1-\alpha)f_I$. This operation means the new meshes' movement is between the moving meshes' movement from DMM and uniform meshes. The results are shown in the table below, illustrating the monotonic relationship between std, range and MM-PDE's loss.
> >
> > | $\alpha$ | std    | range  | Error        |
> > | -------- | ------ | ------ | ------------ |
> > | 0.8      | 0.0581 | 0.3432 | **6.154e-7** |
> > | 0.6      | 0.0708 | 0.4685 | 8.714e-7     |
> > | 0.4      | 0.0826 | 0.5695 | 9.209e-7     |
> > | 0.3      | 0.0939 | 0.6636 | 1.224e-6     |
> >
> >
> >
> > > **Suggestion of taking Geo-FNO as a baseline.**
> >
> > **Answer**: Thanks for your suggestion. According to your requirements, we have added a new baseline Geo-FNO on the flow around a cylinder. New results of this experiment are in the table below. Our MM-PDE still performs the best. This has been added to Section 6.2.
> >
> > | Model | MM-PDE     | GNN    | bigger GNN | CNN  | FNO  | LAMP   | Geo-FNO |
> > | ----- | ---------- | ------ | ---------- | ---- | ---- | ------ | ------- |
> > | Error | **0.0846** | 0.2892 | 0.1548     | -    | -    | 0.4040 | 0.2844  |
> >
> >
> >
> > > **Question about the results of the flow around a cylinder.**
> >
> > **Answer**: Thanks for your question.
> >
> > - Firstly, the error is computed on the horizontal velocity. We have revised Section 6.2 to make it clear.
> > - Secondly, we also compute the relative MSE as below. We can see that the RMSE is relatively low and MM-PDE still surpasses a lot.  This has been added to Appendix E.1.
> >
> > | Model | MM-PDE | GNN    | bigger GNN | LAMP   | Geo-FNO |
> > | ----- | ------ | ------ | ---------- | ------ | ------- |
> > | Error | 0.0197 | 0.0649 | 0.349      | 0.0693 | 0.0602  |
> >
> > - Thirdly, the visualization of the rollout results of this experiment has been added to Appendix E.1's Figure 8, showing that the data produced by MM-PDE aligns well with the real trajectory.
> >
> >
> > > **Question about the cell volumes after mm+end2end training.**
> >
> > **Answer**: Thanks for your question. We find that after mm+end2end training, the std and range of cell volumes under the monitor function we calculated before are not lower. But we also obtain that the monitor function calculated now is not very accurate, since we only get the discrete value of $u$ on the grids and we take the finite difference method to compute the derivatives in the monitor function.
> >
> >
> >
> > > **Question about the way of computing derivatives in DMM loss.**
> >
> > **Answer**: Thanks for your question. We only use the finite difference method to compute the monitor function, as we only obtain the discrete $u$. Then, the other derivatives in physics loss $l$ are all computes using autodifferentiability such as $\frac{\partial \Phi}{\partial x}$ and so on. This explanation has been added to the **Physics loss** part in Section 5.1.
> >
> >
> >
> > > **Question about 3d problems.**
> >
> > **Answer**: Thanks for your constructive question. We have conducted a new experiment on 3d GS reaction-diffusion equation to show that there is no difficulty for our method to extend to higher dimensions. Results in the below table indicate that our proposed method still works in 3d cases. This has been added to Section 6.3.
> >
> > | Model | MM-PDE       | GNN      |
> > | ----- | ------------ | -------- |
> > | MSE   | **1.852e-4** | 2.106e-4 |

---

> ### Comment · Reviewer_W4Uo · 2023-11-21
>
> Thank you to the authors for the extensive reply and the additional runtimes and experiments, especially adding GeoFNO as a strong recent baseline, and even an additional 3D experiment, although the improvements are less impressive in 3D, nevertheless the proposed framework shows it can outperform the baseline. The quantitative evaluation of MM-PDE is much appreciated, yet it does raise some concern on the overhead for using the approach. The extra results on Adam and BFGS helps understand the reasoning for using both. Even though the improvement is incremental, and it is more fine-tuning, it can be seen as an engineering solution to make the framework slightly better.
>
> My questions have been addressed properly, thank you once again to the authors for their time to making the paper stronger by incorporating the reviewer feedback.

---

> > ### Author Response · Authors · 2023-11-22
> > **Response to Reviewer W4Uo**
> >
> > Thank you for your positive assessment of our work.
> >
> > > **Performance of the 3D experiment.**
> >
> > **Answer**: Due to time constraints, we only aim to preliminarily demonstrate that our method can be applied to 3D problems without any issues, without necessarily fine-tuning hyperparameters, such as the crucial hyperparameters of the monitor function and the hyperparameters of GNN. Therefore, the results only show an improvement, although not a significant one.
> >
> > > **Results on Adam and BFGS.**
> >
> > **Answer**: The selection of Adam and BFGS as optimizers is not our main contribution. Indeed, as you have mentioned, these are merely details in the implementation.
> >
> > Again, thanks for your discussion and suggestions!

---

### Official Review · Reviewer_qbBA · 2023-10-31

**Soundness:** 3 good
**Presentation:** 3 good
**Contribution:** 3 good
**Rating:** 6
**Confidence:** 4

**Summary:**

The authors have introduced the Data-free Mesh Mover (DMM) as an innovative component for neural PDE solvers. DMM functions as a mesh adaptation solver, facilitating the adjustment of node positions within a uniform mesh. It's referred to as "data-free" because it achieves an optimal mesh configuration by solving the Monge-Ampere equation. The DMM is subsequently integrated with the MM-PDE (Message passing neural PDE solvers) to enhance overall accuracy.

**Strengths:**

The application of adaptive meshes will improve the accuracy.

**Weaknesses:**

- The efficiency of DMM requires further comprehensive demonstration.
- The contribution of this work appears to be incremental, as the core neural PDE solver used is essentially the message passing neural PDE solver developed by Brandstetter et al.

**Questions:**

- My main concern is on the efficiency of DMM. As data-free, for each input solution $u$, training is needed. Then this can be very costly. Can authors compare the time if using the conventional PDE solver for the mesh adaptation?
- For time-dependent problems, what is the input $u$? At the current time step or the next time step? The singularity of the solution could move. The mesh obtained for $u_k$ may not be suitable for $u_{k+1}$.
- The error listed in Table 1 shows MM-PDE only improves from around $2 \times 10^{-6}$ to $6.7 \times 10^{-7}$ and in Table 2 is from $0.0258$ to $0.0141$ using a bigger GNN. Then a fair comparison is the computational time of MM-PDE and the bigger GNN to achieve the same error. A bigger GNN may need a larger model size, but since no DMM solver is involved, the overall time might be smaller. Again, this is related to Question 1: the efficiency of DMM.
- Evaluation metric. Why not compute the interpolation error directly instead of an upper bound on the interpolation error?
- "Two-order" should be corrected to "second order."

---

> ### Author Response · Authors · 2023-11-20
> **Response to Reviewer qbBA [Part 1]**
>
> We greatly appreciate the suggestions from the reviewers, which have enabled us to further enhance our manuscript.
>
>
> > **Concerns about the efficiency of DMM.**
>
> **Answer**: Thanks for your question. To make it clear, we have revised Section 6.1 to show that the Data-free Mesh Mover (DMM) is fast in inference.
>
> - The DMM is an operator trained on data belonging to a distribution rather than a single state. Therefore, once training is complete, it requires only a brief inference time when applied to new data from the same distribution.
>
> - Regarding DMM: we have calculated the inference time of the DMM to be 0.001s for the Burgers' equation and 0.088s for the flow around a cylinder, whereas the conventional PDE solver for the Burgers' equation's mesh adaptation requires several tens of seconds [1].
>
> - Regarding MM-PDE: when we use MM-PDE, the extra cost brought by DMM is comparable with the inference time. A comparison of the MMPDE's time with other baselines' is provided in the table below, showing that its time is comparable to other models. And we have added these above results to the new version of our paper in the section of the 'Burgers' equation'.
>
> | Model   | MM-PDE (MP-PDE) | MP-PDE | bigger MO-PDE | MM-PDE (MGN) | MGN    | CNN    | FNO    | LAMP   |
> | ------- | --------------- | ------ | ------------- | ------------ | ------ | ------ | ------ | ------ |
> | Time(s) | 0.5192          | 0.3078 | 0.3298        | 0.1187       | 0.0400 | 0.0027 | 0.0159 | 1.4598 |
>
> [1] Song W, Zhang M, Wallwork J G, et al. M2N: mesh movement networks for PDE solvers[J]. Advances in Neural Information Processing Systems, 2022, 35: 7199-7210.
>
>
>
> > **Concerns for the core contribution.**
>
> **Answer**: Thanks for your discussion about the contribution. Please note that our contribution includes design on both DMM and MM-PDE rather than choosing a specific neural network structure for MM-PDE.
>
> - To further demonstrate that our contribution is orthogonal to the neural network structure in MM-PDE, we have conducted an additional experiment using another model architecture Mesh Graph Nets commonly used to handle irregular grids [1]. This is to showcase the universality and novelty of our method. The table below shows the experimental results of the Burgers' equation. From the table, the error of MGN combined with the MM-PDE framework still surpasses MGN, which indicates the effectiveness of our method. The results are added to the 'Burgers' equation' section.
>
> | Model | MM-PDE (MGN) | MGN      |
> | ----- | ------------ | -------- |
> | Error | **2.138e-5** | 2.459e-5 |
>
> - For clarity, we have revised our conclusion section to make our contribution clearer: As for DMM, we introduce the physics loss to make it data-free, and we design the model architecture, sampling strategy, and training framework to enhance performance. As for the MM-PDE, we propose a framework to improve the base model and have added many design features tailored specifically for the moving meshes, such as the two-branch architecture, the residual cut network, and interpolation networks capable of adaptively generating interpolation weights. Given the dynamic and irregular grid, we thus choose the GNN network structure employed by Brandstetter et al. as our base architecture.
>
> [1] Pfaff T, Fortunato M, Sanchez-Gonzalez A, et al. Learning mesh-based simulation with graph networks[J]. arXiv preprint arXiv:2010.03409, 2020.
>
>
>
> > **Question about only using the current state for moving mesh.**
>
> **Answer**: Thank you for this question. In practice, we use the state of the current time step as input.
>
> - On the one hand, PDE solvers can only see the state of the current time step during the solving process.
> - On the other hand, to the best of our knowledge, traditional methods that use r-adaptive meshes also generate adaptive meshes based on the current time step [1, 2, 3].
> - However, we also considered the issue you raised before. We may be able to give the mesh mover the ability to predict the future by introducing a surrogate model to provide information about the future. We had originally intended to explore this idea as a topic for future work. Now we have incorporated our thoughts on this issue into the 'Conclusion and Discussion' section of the manuscript.
>
> [1] Budd C J, Cullen M J P, Walsh E J. Monge–Ampére based moving mesh methods for numerical weather prediction, with applications to the Eady problem[J]. Journal of Computational Physics, 2013, 236: 247-270.
>
> [2] Wallwork J G, Barral N, Ham D A, et al. Goal-oriented error estimation and mesh adaptation for tracer transport modelling[J]. Computer-Aided Design, 2022, 145: 103187.
>
> [3] Budd C J, Williams J F. Parabolic Monge–Ampère methods for blow-up problems in several spatial dimensions[J]. Journal of Physics A: Mathematical and General, 2006, 39(19): 5425.

---

> > ### Author Response · Authors · 2023-11-20
> > **Response to Reviewer qbBA [Part 2]**
> >
> > > **Question about considering the interpolation error bound instead of interpolation error.**
> >
> > **Answer**: Thanks for your question. Firstly, it is a commonly considered setting [1, 2, 3]. Secondly, we consider the interpolation error in the Sobolev space rather than a specific state *u*, so we deduce the interpolation error bound instead of a specific interpolation error value. We have added an explanation in the second paragraph of Section 4.
> >
> > [1] Huang W, Russell R D. Adaptive moving mesh methods[M]. Springer Science & Business Media, 2010.
> >
> > [2] Hetmaniuk U, Knupp P. An R-adaptive Mesh Optimization Algorithm Based on Interpolation Error Bounds[R]. Sandia National Lab. (SNL-NM), Albuquerque, NM (United States), 2007
> >
> > [3] Moxey D, Sastry S P, Kirby R M. Interpolation error bounds for curvilinear finite elements and their implications on adaptive mesh refinement[J]. Journal of Scientific Computing, 2019, 78(2): 1045-1062.
> >
> >
> >
> > > **Question about the typos.**
> >
> > **Answer**: Thank you for raising this issue. We have revised the entire paper to fix the typos and enhance its readability.

---

> > > ### Comment · Reviewer_qbBA · 2023-11-22
> > >
> > > The authors have successfully addressed my primary concern regarding the efficiency of DMM, particularly noting that the interference time is within an acceptable range. This positive development prompts me to consider raising my score. However, I still find the improvement of MM-PDE over GNN to be less impressive than expected.

---

> > > > ### Author Response · Authors · 2023-11-22
> > > > **Response to Reviewer qbBA**
> > > >
> > > > Thank you very much for your constructive comments and happy to know that we have successfully addressed your concerns. Besides, we would like to defend the significance of our contribution as follows:
> > > >
> > > > 1. Based on our research, we are the first to implement physics-informed, data-free moving mesh operator learning. Compared to a series of methods in related work based on supervised learning and reinforcement learning, our method does not rely on moving mesh data. Also, by introducing physical information, we have improved the generalizability of the model.
> > > > 2. We are also pioneers in exploring the utilization of moving meshes to facilitate the learning of neural PDE solvers. This is a crucial topic, yet previous related studies have not ventured deeply into this area. They have only applied moving meshes to traditional PDE solvers.
> > > > 3. The aforementioned two points have been discussed in detail in both the introduction section and the related work section.
> > > > 4. Regarding the improvement of GNN you mentioned, it is actually not the starting point of our work, nor is it our core contribution. It is merely an experimental demonstration that we can successfully perform moving mesh using the physical information of the MA equation, and this moving mesh can indeed help the learning of neural PDE solvers. By utilizing our two core contributions, we can delve deeper into the future to understand how moving meshes can have a more profound impact on the structure, loss function, etc., of the learning of neural PDE solvers.
> > > >
> > > > Hope to hear more comments from you, and thank you again for your efforts to significantly improve the quality of the paper.

---

### Author Response · Authors · 2023-11-20
**General Comments and Revision Summary**

We express our sincere gratitude to all reviewers for the careful review, constructive suggestions and questions. We also appreciate the recognition given to our work's insights, ideas and rich theory. The primary revisions made to the paper are summarized below.

- According to reviewers' suggestions and questions, we have conducted new experiments including a new baseline Geo-FNO, MM-PDE combined with another GNN PDE solver Mesh Graph Net, a 3-D main experiment, and six ablation studies. The new ablation studies contain experiments regarding to physics loss optimization, resolution transformation, mesh tangling and $l_{convex}$, roles of three losses within the physics loss, relationship between std, range and MM-PDE, and MM-PDE with MGN.
- We have revised the presentation of the paper and added explanations and details to make the paper clearer, such as the reason to consider the interpolation bound instead of the interpolation error itself, the way of calculating derivatives in the physics loss, and why we take BFGS to optimize the last layer's parameters
- Due to the concerns about the efficiency of DMM raised by reviewers, we have provided the memory, number of parameters, training time and inference time of DMM. We have also reported the inference time of MM-PDE and other baselines.
- We have added new discussions about mesh tangling, the input state of the monitor function, and cases where the monitor function gives large values near boundaries.
- We have provided new visualization and relative MSE to help readers understand the result intuitively. The figures include $48\times48$ meshes and rollout results of flow around a cylinder.
- Last but not least, we thank all reviewers again for their efforts that enhance our paper.

---

### Meta-Review · Area_Chair_JzoR · 2023-12-10

**Metareview:**

This paper seems to be a typical "borderline" case, with several strengths on the conceptual side, but also several weaknesses regarding depth/significance of the contribution and experimental improvements over related approaches.  After the rebuttal and discussion phase, I had the impression that some of the concerns could be addressed by the authors, but that there are still some open questions regarding the significance/relevance of the contribution. Given the fact that the strengths of the paper lie on the conceptual side, and potential weaknesses mainly on the experimental side, for me the positive aspects (slightly) outweigh the negative ones. Therefore I recommend (weak) acceptance.

**Justification For Why Not Higher Score:**

There are too many open questions regarding the impact of this work.

**Justification For Why Not Lower Score:**

The paper contains some nice conceptual ideas which might be interesting to the research community.

---

### Decision · Program_Chairs · 2024-01-16

Accept (poster)